# Ratiometric Matryoshka biosensors from a nested cassette of green- and orange-emitting fluorescent proteins

Cindy Ast[1], Jessica Foret[1], Luke M. Oltrogge[2,5], Roberto De Michele[3], Thomas J. Kleist[1], Cheng-Hsun Ho[1,4] & Wolf B. Frommer[1]

Sensitivity, dynamic and detection range as well as exclusion of expression and instrumental artifacts are critical for the quantitation of data obtained with fluorescent protein (FP)-based biosensors in vivo. Current biosensors designs are, in general, unable to simultaneously meet all these criteria. Here, we describe a generalizable platform to create dual-FP biosensors with large dynamic ranges by employing a single FP-cassette, named GO-(Green-Orange) Matryoshka. The cassette nests a stable reference FP (large Stokes shift LSSmOrange) within a reporter FP (circularly permuted green FP). GO- Matryoshka yields green and orange fluorescence upon blue excitation. As proof of concept, we converted existing, single-emission biosensors into a series of ratiometric calcium sensors (MatryoshCaMP6s) and ammonium transport activity sensors (AmTryoshka1;3). We additionally identified the internal acid-base equilibrium as a key determinant of the GCaMP dynamic range. Matryoshka technology promises flexibility in the design of a wide spectrum of ratiometric biosensors and expanded in vivo applications.

[1] Department of Plant Biology, Carnegie Science, Stanford, California 94305, USA. [2] Department of Chemistry, Stanford University, Stanford, California 94305, USA. [3] Institute of Biosciences and Bioresources (IBBR), Italian National Research Council (CNR), Palermo 90129, Italy. [4] Agricultural Biotechnology Research Center, Academia Sinica, Academia Road, Nankang, Taipei, Taiwan. [5] Present address: Department of Molecular and Cell Biology, University of California, Berkeley, Califonia 94720, USA. Correspondence and requests for materials should be addressed to W.B.F. (email: frommew@hhu.de)

Presently, genetically encoded biosensors fall into two categories: single-fluorescent protein (FP) sensors and dual-FP sensors. Single-FP sensors either exploit the intrinsic sensitivities of the FP itself to certain stimuli, such as pH[1], or the FP is fused to a recognition element that is sensitive to a specific analyte or process[2]. Alternatively, the FP can be sensitized to conformational changes of an attached recognition element by circular permutation (cpFP)[3]. Most dual-FP biosensors consist of a recognition domain sandwiched between two FPs with properties that allow for FRET (Förster Resonance Energy Transfer; the non-radiative energy transfer between a donor and an acceptor FP)[2, 4, 5]. Conformational rearrangements in the recognition domain affect the relative intensity of the two FPs and, therefore, the ratiometric readout.

Both single-FP and FRET biosensors are characterized by specific advantages and limitations. Single-FP biosensors can achieve a large dynamic and wide detection range, and a high signal-to-noise ratio, thus creating great sensitivity. Most single-FP biosensors are intensiometric, as they rely on the readout of a single fluorescence intensity (FI). However, detecting changes only in a single emission range presents a disadvantage, as the signal is prone to artifacts, such as changes in expression level of the sensor or instrumental effects, due to bleaching and motion, and may thus lead to misinterpretation of the data. Therefore, due to the lack of an internal reference, single-FP biosensors do not provide absolute quantitative information. This is especially crucial during the screening of a random single-FP biosensor library, where quantitation is essential to correct for expression artifacts[6–9]. This drawback is commonly addressed by co-expression or terminal fusion of a spectrally distinct FP, such as mCherry, as a normalization control[6–12]. However, co-expression of the reference is not a reliably accurate control, as for example, variation in expression or protein levels can lead to artifacts of the signal readout. Alternatively, ratiometric single-FP biosensors have been developed, exploiting the two distinct absorbance maxima of the protonated and deprotonated FP-chromophore. The resulting biosensors display two excitation or emission maxima with opposite intensity changes[13–17]. However, these sensors rely on excited-state proton transfer from the protonated FP-chromophore, requiring an intact proton network and near-ultraviolet excitation[18]. In biological systems, ultraviolet radiation can lead to photodamage and is thus problematic[19]. In contrast, FRET biosensors intrinsically provide a ratiometric readout and contain a reference, as the energy accepting FP can be independently excited and monitored. However, FRET biosensors face three limitations. Firstly, except for rare cases, the detection range of FRET biosensors is limited to two orders of magnitude of analyte concentration[20]. Secondly, FRET sensors are restricted in their dynamic range due to the size of the FP-barrel, which limits the proximity of the chromophores[21]. Thirdly, the coupling of the fluorophores via rotatable and flexible peptide linkers leads to signal loss due to rotational averaging[22].

The design of biosensors is still largely empirical, often requiring the analysis of a large number of constructs[23]. We therefore aimed to develop a generalizable platform for rapid engineering of ratiometric biosensors that retain the features of single-FP biosensors but contain an internal reference FP. The peptide loop created by circular permutation of FPs (cpFP), tolerates nested insertion of a second FP[3]. While the cpFP serves as the reporter, the second, nested FP serves as the internal control. The two green cpFP variants tested here, an enhanced cpEGFP[24, 25] and a superfolder cpsfGFP[26], both tolerated the nested reference FP, a large Stokes shift (LSS) mOrange[27, 28]. The LSS and the excitation overlap of LSSmOrange with the green cpFP allowed for single excitation at $\lambda_{exc}$ 440 nm yielding green and orange emissions, thus providing a ratiometric readout. At the same time, the emission spectrum of the green cpFP shows little spectral overlap with the absorption spectrum of LSSmOrange, therefore limiting FRET. FRET compromises the ratiometric readout, as evidenced by an alternative construct generated with CyOFP1 as nested reference[29]. The effects of FRET on the sensor properties were also demonstrated by the recently generated ratiometric tandem-fusions called GCaMP-Rs[12].

We hypothesized that insertion of the nested FPs into suitable positions of a recognition element would allow for creation of ratiometric biosensors in a single cloning step. Because this concept is reminiscent of the nested Russian dolls, we named this technology 'Matryoshka' [mä′trē-ō′shkə].

As proof of concept, we used the Matryoshka technology to engineer ratiometric calcium sensors based on GCaMP6s, the slow version of GCaMP6[6]. The resulting MatryoshCaMP6s variants retained the high dynamic range of GCaMP6s without substantial effects on other in vitro properties (steady-state fluorescence spectra, $K_d$ and p$K_a$) relative to the parent sensors. We further demonstrated their suitability for in vivo applications using mammalian HEK cells and stably transformed Arabidopsis seedlings.

The different dynamic ranges of the calcium sensor iterations prompted us to analyze the calcium sensor responses in quantitative detail. Careful analysis of the data and mathematical modeling led us to discover a major and previously undescribed contributor to the dynamic range arising from the internal FP acid-base equilibrium. This factor is independent of the pH and constitutes the single largest contribution to the dynamic range of GCaMP6s.

Furthermore, we demonstrated the suitability of the technology for membrane transporter proteins by engineering and deploying Matryoshka-type transport activity sensors based on AmTrac[25], named AmTryoshka1;3. In living yeast cells, the best performing AmTryoshka1;3 mutant reported a FI change of 30% in the green emission channel, while the orange channel remained stable.

Our results demonstrate that Matryoshka is a promising technology for engineering ratiometric biosensors that contain a stoichiometric reference FP as internal standard for improved analyte quantification. GO-Matryoshka combines the reporter FP and the reference FP in a single cassette, which can be inserted into a recognition element of interest in a single cloning step, simplifying sensor construction and analysis. The Matryoshka-based biosensors use a single excitation wavelength to report the relative emission of two FPs. Additionally, our mathematical model may provide for a more refined understanding of biosensor dynamic range and the identification of the residues that affect the acid-base equilibrium, which may facilitate future sensor design and optimization.

## Results

**GO-Matryoshka cassette for designing ratiometric biosensors.** To design a tool suitable for one-step generation of ratiometric biosensors with wide dynamic ranges, an FP with a large Stokes shift, LSSmOrange, was inserted as a stable reference domain into a green cpFP. Two different variants of cpFPs were tested as reporters: cpEGFP, which has been used to construct GCaMP and other single-FP sensors[14, 24, 25, 30], and cpsfGFP, a variant with improved brightness and folding properties compared to cpEGFP[26]. LSSmOrange was inserted into the center of the GGT-GGS sequence, which connects the N- and C-termini of both the original EGFP[3] and its variant sfGFP[31, 32]. We named the resulting dual-FP combinations eGO-Matryoshka (based on cpEGFP) and sfGO-Matryoshka (based on cpsfGFP; Fig. 1a).

The residues that flank the cpFP at its N and C termini, engineered during the process of circular permutation, have been

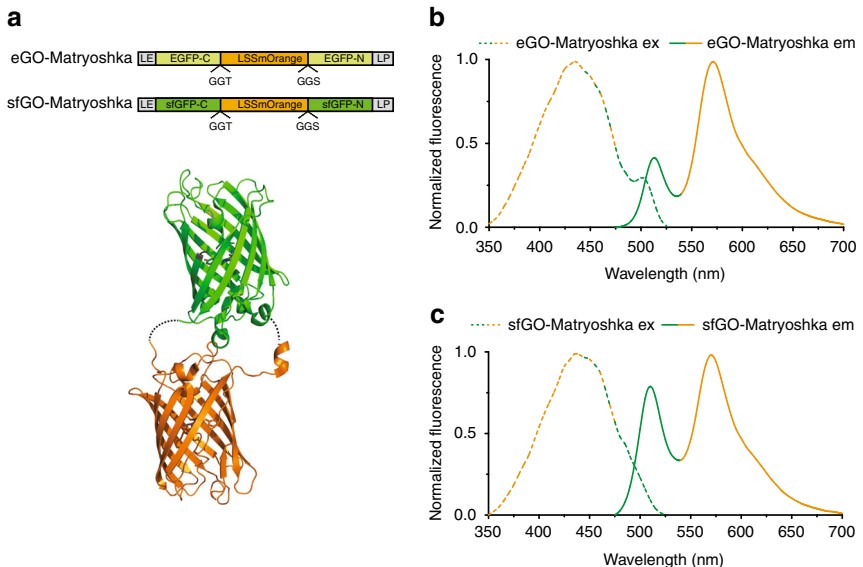

**Fig. 1** GO-Matryoshka design. **a** Schematic representation of eGO-Matryoshka and sfGO-Matryoshka: LSSmOrange (reference domain) is sandwiched between the C and N termini of EGFP (*light green*) and sfGFP (*dark green*). LE and LP (*dark gray*) represent the flanking residues. **b**, **c** Normalized steady-state fluorescence excitation ($\lambda_{em}$ 570 nm; *dashed line*) and emission ($\lambda_{exc}$ 440 nm; *solid line*) of eGO-Matryoshka (**b**) and sfGO-Matryoshka (**c**) at pH 10.5 (FI is maximal at this pH; independent spectral analysis of four biological replicates). Note that the intensities in the green emission channel are affected by the terminal residues, as they affect the cpFP FI, chromophore quantum yield and p$K_a$. The combination LS/FN yielded a more acidic p$K_a$ than the LE/LP combination (Supplementary Fig. 2)

reported to affect the protonation equilibrium of the chromophore and, thus, the fluorescence properties of the cpFP[25]. They connect the sensor domain with the seventh β-strand of the cpFP. The seventh β-strand interacts with the FP-chromophore, and can thus affect the dynamic range and FI of the sensor[26, 31]. Therefore, the flanking residues were maintained throughout the GO-Matryoshka characterization. The combination used, an N-terminal amino acid pair of leucine/glutamic acid (LE) and a C-terminal amino acid pair of leucine/proline (LP), had shown the best performance for GCaMP6[6].

In vitro characterization of the purified GO-Matryoshka iterations revealed two emission maxima, at $\lambda_{em}$ ~ 510 nm and at $\lambda_{em}$ ~ 570 nm, respectively, upon excitation at $\lambda_{exc}$ 440 nm (Fig. 1b, c). Green emission with minimal or no cross-excitation of LSSmOrange at $\lambda_{exc}$ 485 nm was observed (Supplementary Fig. 1a). Comparative analysis of the steady-state spectral properties of the individual FPs revealed that eGO-Matryoshka and sfGO-Matryoshka exhibited no detectable changes in excitation or emission maxima compared to LSSmOrange and cpEGFP or cpsfGFP, respectively (Supplementary Fig. 1b).

Circularly permuted FPs are intrinsically sensitive to conformational changes, and alterations of the overall cpFP structure may result in modulations of the chromophore p$K_a$. To test whether the cpFPs were affected by the LSSmOrange insertion, we performed pH titrations. The resulting p$K_a$ values were 8.2–8.3 for the cpFP (cpEGFP or cpsfGFP) and the GO-Matryoshka iterations (eGO-Matryoshka and sfGO-Matryoshka; Supplementary Fig. 2, Supplementary Table 1) upon excitation at $\lambda_{exc}$ 440 or 485 nm.

Nesting of the FPs may potentially result in reduced maturation efficiencies of the FP-chromophores. To assess if the chromophore maturation is compromised in the Matryoshka iterations, we analyzed the individual and nested FPs by optical absorbance spectroscopy and intact protein mass spectrometry (MS; Supplementary Fig. 3, Supplementary Note 1). The absorbance data indicate reduced ratios of LSSmOrange to cpFP of 0.59 for eGO-Matryoshka and 0.82 for sfGO-Matryoshka

(Supplementary Fig. 3a, b), suggesting that the LSSmOrange chromophore does not reach complete maturation, as opposed to the cpFP chromophores. However, our MS data analysis indicates that the reduced LSSmOrange maturation efficiency in the GO-Matryoshkas is very similar to LSSmOrange alone and that the cpFPs do not influence the extent of maturation of the LSSmOrange (Supplementary Fig. 3c–e). Therefore, we infer that the identified incomplete maturation of LSSmOrange in the GO-Matryoshkas is likely not a consequence of the nesting approach but a property of the LSSmOrange alone, which is not uncommon for red FPs[33]. Additionally, we found the fraction of immature LSSmOrange to be consistent and should therefore not pose a limitation for the Matryoshka concept. However, the obtained ratios should not be treated as fixed values as they were obtained from purified protein from *E.coli* and might differ depending on which biological system is used.

We conclude that insertion of LSSmOrange did not lead to detectable alterations in the steady-state properties, p$K_a$ values and maturation efficiencies of the individual FPs compared to the GO-Matryoshka versions. The distinct green and orange emission maxima allow for spectral separation of both bands and for ratiometric readout. Both GO-Matryoshka cassettes tested as suitable tools for ratiometric biosensor design.

**Generation of MatryoshCaMP6s calcium sensors.** GCaMP6s is among the most sensitive calcium sensors[6] and therefore was chosen to test the Matryoshka technology. GCaMP6s carries cpEGFP between a calcium-binding calmodulin (CaM) domain and a CaM-interacting M13 peptide. Numerous mutations have been found that further increase the sensitivity of the sensors, i.e. residue mutations at the interface between cpEGFP and CaM or a K78H mutation in the cpEGFP portion[6]. Since the K78H (amino acid numbering according to[6]) mutation has been implicated in increased sensitivity of GCaMP6s relative to GCaMP5G and other GCaMP6 sensors[6, 34], its effect on cpsfGFP-based sensors was also explored. In cpsfGFP, the residue equivalent to H78 is a

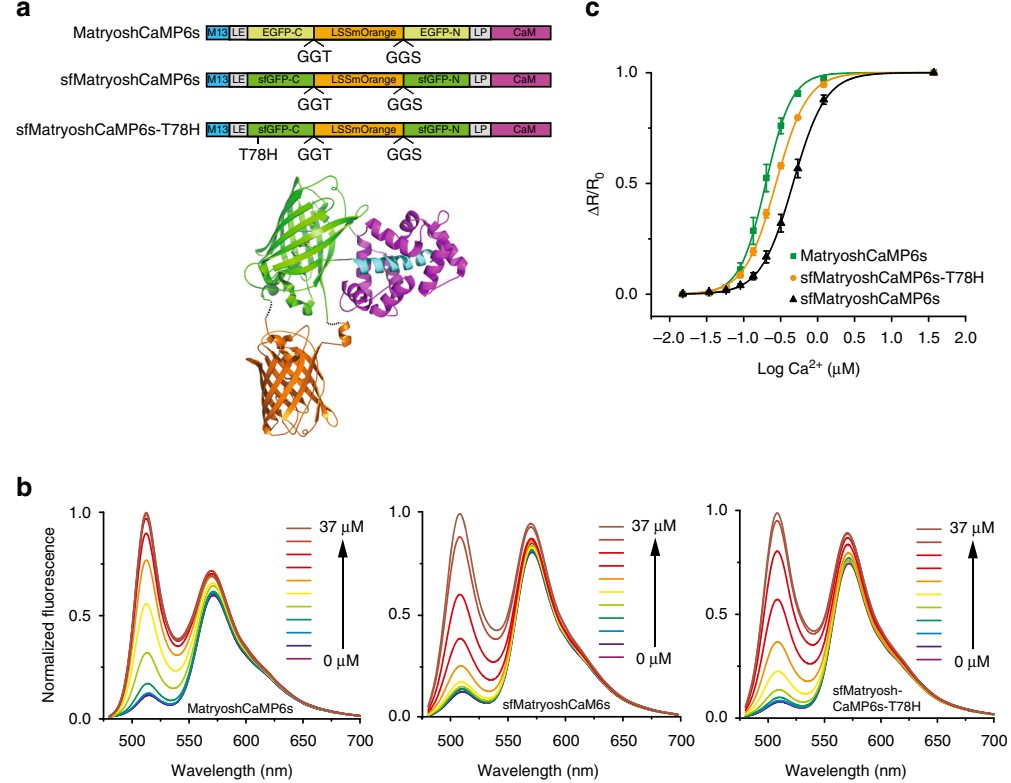

**Fig. 2** In vitro characterization of MatryoshCaMP6s and sfMatryoshCaMP6s and sfMatryoshCaMP6s-T78H. **a** Schematic representation of MatryoshCaMP6s sensors, composed of GO-Matryoshka (LSSmOrange sandwiched between the C and N termini of either EGFP, sfGFP or sfGFP-T78H) inserted between the M13 peptide and calmodulin domain. LE and LP represent flanking residues that connect the cpFP domains with the recognition domains. **b** Steady-state fluorescence spectra ($\lambda_{exc}$ 440 nm) of sensors titrated with free calcium (0, 0.02, 0.03, 0.06, 0.09, 0.13, 0.20, 0.31, 0.52, 1.2 and 37 $\mu$M). The FI change in the orange channel mainly derives from fluorescence bleed through. **c** Calcium-affinity titrations (R = FI$_{510nm}$/FI$_{570nm}$) corresponding to the spectra in **b**. Data were corrected for fluorescence bleed-through (factor 0.1; mean ± s.e.m. from biological replicates with $n = 6$ for MatryoshCaMP6s and sfMatryoshCaMP6s, $n = 4$ for sfMatryoshCaMP6s-T78H)

**Table 1 In vitro properties of purified GCaMP6 and MatryoshCaMP6s calcium sensors**

| Sensor | $K_d$ (nM) $\Delta F/F_0$ | $K_d$ (nM) $\Delta R/R_0$ | $pK_{a,apo}$ | $pK_{a,sat}$ | Dynamic range$_{440\ exc}$ $\Delta R/R_0$ | Dynamic range$_{440\ exc}$ $\Delta F/F_0$ | Dynamic range$_{485\ exc}$ $\Delta F/F_0$ |
|---|---|---|---|---|---|---|---|
| GCaMP6s | 175 ± 17 | | ~ 8[a] | 6.13 ± 0.03 | | 9.8 ± 0.3 | 49.7 ± 0.4 |
| MatryoshCaMP6s | 197 ± 23 | 196 ± 22 | ~ 8[a] | 6.09 ± 0.03 | 8.5 ± 0.2 | 8.6 ± 0.1 | 41.8 ± 0.9 |
| sfGCaMP6s | 481 ± 45 | | 8.26 ± 0.01 | 6.01 ± 0.03 | | 9.5 ± 1.1 | 12.7 ± 0.2 |
| sfMatryoshCaMP6s | 501 ± 64 | 483 ± 51 | 8.14 ± 0.01 | 6.03 ± 0.03 | 7.6 ± 0.3 | 8.1 ± 0.5 | 9.1 ± 0.4 |
| sfGCaMP6s-T78H | 303 ± 28 | | 8.47 ± 0.01 | 5.78 ± 0.02 | | 11.4 ± 0.5 | 19.3 ± 2.8 |
| sfMatryoshCaMP6s-T78H | 271 ± 10 | 265 ± 8 | 8.34 ± 0.02 | 5.74 ± 0.03 | 11.9 ± 0.6 | 12.1 ± 0.5 | 16.4 ± 0.8 |

Calcium affinities and pKa values retrieved from the calcium and pH titration fits in Fig. 2c and Supplementary Fig. 5, respectively
[a]Values are estimates, since saturation was not reached at the maximal pH tested here (pH 10.5)

threonine (T78), thus a T78H variant of cpsfGFP-based sensors was analyzed.

To be able to compare sensors containing different cpFP variants (cpEGFP, cpsfGFP and cpsfGFP-T78H), three Matryoshka iterations coupled to the GCaMP6s recognition elements were generated: (1) LSSmOrange was inserted into the flexible loop of cpEGFP in GCaMP6s (named MatryoshCaMP6s); (2) cpEGFP was replaced with sfGO-Matryoshka (named sfMatryoshCaMP6s); and (3) T78H was introduced into sfMatryoshCaMP6s (named sfMatryoshCaMP6s-T78H) (Fig. 2a).

In vitro characterization of the purified GO-Matryoshka-coupled calcium sensors revealed two emission maxima, at $\lambda_{em}$ ~ 510 nm and $\lambda_{em}$ ~ 570 nm, when excited at $\lambda_{exc}$ 440 nm (Fig. 2b). Upon $\lambda_{exc}$ 485 nm excitation, a single emission

maximum at $\lambda_{em}$ ~ 510 nm was detected, with minimal cross-excitation of LSSmOrange (Supplementary Fig. 4). Calcium treatment yielded a large positive response in the green emission channel for all sensors (Fig. 2b). Upon calcium addition, a small FI increase was detected in the orange channel. This increase can be attributed to the increased FI from the cpFP as part of the sensor response and was calculated as a ~ 10% fluorescence bleed-through from green emission into the orange emission channel. A fluorescence bleed-through factor of 0.1 was taken into account in subsequent ratiometric analyses of MatryoshCaMP6s data.

Quantitative analysis of the calcium titration revealed specific affinities for the three MatryoshCaMP6s iterations, with different apparent dissociation constants ($K_d$) ranging from 200–500 nM (Fig. 2c, Table 1). MatryoshCaMP6s had the highest

affinity (196 ± 22 nM; mean ± s.e.m. $n = 6$). The affinities of sfMatryoshCaMP6s-T78H and sfMatryoshCaMP6s were 1.4-fold (265 ± 8 nM; mean ± s.e.m. $n = 4$) and 2.5-fold (483 ± 51 nM; mean ± s.e.m. $n = 6$) lower, respectively. Overall, the affinities were comparable to those of the parent sensors (GCaMP6s, sfGCaMP6s and sfGCaMP6s-T78H; Table 1).

GCaMP calcium sensors are pH sensitive, and the $pK_a$ varies among the sensor variants. As described for GCaMP6[6], the $pK_a$ values at saturating calcium conditions ($pK_{a,sat}$) were more acidic compared to the calcium-free conditions ($pK_{a,apo}$). sfGCaMP6s-T78H and sfMatryoshCaMP6s-T78H showed the largest change between $pK_{a,apo}$ and $pK_{a,sat}$, with differences of ~ 2.6 pH units, followed by sfGCaMP6s and sfMatryoshCaMP6s, with differences of ~ 2.2 pH units. GCaMP6s and MatryoshCaMP6s showed the lowest differences, ~ 1.8 pH units, between $pK_{a,apo}$ and $pK_{a,sat}$ (Table 1, Supplementary Fig. 5).

The dynamic range ($\Delta R/R_0$) calculated for the ratiometric MatryoshCaMP6s variants ranged from 7.6 to 12 (Table 1; mean ± s.e.m.). sfMatryoshCaMP6s-T78H showed the highest value (11.9 ± 0.6; $n = 4$), followed by MatryoshCaMP6s (8.5 ± 0.2; $n = 6$) and sfMatryoshCaMP6s (7.6 ± 0.3 $n = 5$). The dynamic range for the parent sensors (GCaMP6s, sfGCaMP6s and sfGCaMP6s-T78H) was estimated using $\Delta F/F_0$ at $\lambda_{exc}$ 440 nm, which revealed values consistent with the $\Delta R/R_0$ (Table 1). The dynamic ranges were also estimated using $\Delta F/F_0$ at $\lambda_{exc}$ 485 nm. As expected, calculation of the dynamic range using $\Delta F/F_0$ at $\lambda_{exc}$ 485 nm yielded a much larger dynamic range for Matryosh-CaMP6s (41.8 ± 0.9; $n = 6$) and its parent GCaMP6s (49.7 ± 0.4; $n = 6$) than using $\lambda_{exc}$ 440 nm (Table 1; mean ± s.e.m.). The cpsfGFP- or sfGO-Matryoshka-based sensors also retained useful dynamic ranges, with higher values for sfMatryoshCaMP6s-T78H (16.4 ± 0.8; $n = 4$) and sfGCaMP6s-T78H (19.3 ± 0.28; $n = 5$) than for sfMatryoshCaMP6s (9.1 ± 0.4; $n = 5$) and sfGCaMP6s (12.7 ± 0.92; $n = 4$).

The detection range for free calcium, which we defined as $\Delta R/R_0$ at 0.1 and 0.9 ligand occupancy (Fig. 2c), ranged from 0.1–1.3 μM for the MatryoshCaMP6s sensors. sfMatryosh-CaMP6s had the largest detection range (0.2–1.3 μM), followed by sfMatryoshCaMP6s-T78H (0.1–0.8 μM) and Matryosh-CaMP6s (0.1–0.5 μM).

Altogether, the in vitro characterization and evaluation of the different MatryoshCaMP6s sensors at either excitation maxima ($\lambda_{exc}$ 440 and 485 nm) revealed no detectable effects of the LSSmOrange insertion relative to the original GCaMP6s sensors. However, depending on which cpFP variant was used (cpEGFP, cpsfGFP or cpsfGFP-T78H), the calcium-binding affinities, dynamic range and detection range differed.

To better understand the dynamic range differences, we developed a mathematical model that describes the biosensor responses as a function of pH. We went beyond a simple, single-site chromophore titration description and used the more complete inter-site coupling model for FPs model, which emerged over recent years[31, 35, 36]. With experimental pH-dependent absorbance and fluorescence excitation spectra of apo and saturated GCaMP6s and sfGCaMP6s, we were able to para-meterize the model and elucidate the factors responsible for the dynamic range (Supplementary Note 2). Besides the well-established factors of differential $pK_a$'s and quantum yields of the ligand-saturated and apo species, a new factor was identified: the internal acid-base equilibrium. Our analysis indicated that the internal acid-base equilibrium plays a dominant role in establish-ing the large dynamic range for GCaMP6s sensors. sfGCaMP6s, in contrast, derives its response mainly from the differential $pK_a$'s between the apo and saturated species (Supplementary Note 2).

To test whether other FPs are suitable as stable reference FPs, we constructed variants of MatryoshCaMP6s that contained

CyOFP1 instead of LSSmOrange[29]. CyOFP1 is a LSS FP whose excitation spectrum shows larger spectral overlap with GFP excitation but a more red-shifted emission compared to LSSmOrange. CyOFP1 was well tolerated in the resulting CyOFP1-containing MatryoshCaMP6s variants (Supplementary Fig. 6, Supplementary Table 2). However, due to extensive overlap of the emission spectrum of cpEGFP/cpsfGFP with the absorp-tion spectrum of CyOFP1 the occurrence of FRET must be taken into account, which compromises the ratiometric readout. This was demonstrated by ~ 20% reduced calcium affinities and an up to 50% reduced dynamic range when $\Delta R/R_0$ ($\lambda_{exc}$ 485 nm) was evaluated as opposed to $\Delta F/F_0$ ($\lambda_{exc}$ 485 nm; Supplementary Table 2). LSSmOrange-based MatryoshCaMP6s iterations on the contrary do not show obvious changes in calcium affinities when analyzed for $\Delta R/R_0$ ($\lambda_{exc}$ 440 nm) vs. $\Delta F/F_0$ ($\lambda_{exc}$ 485 nm) (Table 1). This is due to no or only minimal FRET as evidenced by the minimal overlap of donor emission and acceptor absorbance (Supplementary Fig. 1b) and the lack of fluorescence lifetime changes of the cpEGFP and cpsfGFP moiety (data not shown). Thus, subsequent in vivo analyses focused on the MatryoshCaMP6s iterations carrying the LSSmOrange.

**MatryoshCaMP6s reports cytosolic calcium elevation in plants.** Plants respond to diverse forms of environmental stimuli with transient rises in cytosolic free calcium levels. Because exposure to salt (NaCl) stress is a thoroughly documented elicitor of calcium transients in plants[37, 38] we used this treatment to validate that MatryoshCaMP6s is suitable for monitoring cytosolic calcium levels in intact plants. Roots of one-week old seedlings expressing MatryoshCaMP6s were exposed to salt shock by addition of NaCl in liquid media (final concentration ~ 50 mM; Fig. 3a). The treatment evoked a rapid and pronounced FI change in the green emission channel, i.e., mean pixel intensity values were elevated by ~ 96%, whereas the orange emission channel remained com-paratively stable (Fig. 3b, c, Supplementary Fig. 7a), i.e., mean pixel intensity values were elevated by ~ 10%, similar to the bleed-through factor determined in vitro. Effects from background fluorescence can be excluded since the untransformed control seedling did not show detectable autofluorescence or an obser-vable ratio change (Fig. 3c, Supplementary Fig. 7b). In response to salt shock treatment, fluorescence monitoring indicated that cytosolic calcium levels increased rapidly (within 4 s of treat-ment), reached peak intensity in about 39 s, and returned to baseline conditions after about 108 s of the peak response without removal of the salt stress (Fig. 3, Supplementary Movies 1–4). Effects of pH can likely be excluded, since an increase in cytosolic calcium has been reported to lead to cytosolic acidification[39], which would lead to reduced FI as opposed to the FI increase we observe (Supplementary Fig. 5). Thus, these data show that MatryoshCaMP6s is a suitable tool for monitoring calcium transients in intact plants.

**Mammalian cell assays using MatryoshCaMP6s.** In addition to monitoring calcium signals in plants, we sought to demonstrate the suitability of MatryoshCaMP6s in other eukaryotic systems. To evaluate the suitability of the MatryoshCaMP6s variants for future applications in mammalian systems, we expressed a selection of the sensors (GCaMP6s or MatryoshCaMP6s) in HEK293T cells and induced calcium spikes using methacholine (MeCh) as described[40]. As expected, the green FI at 440 nm laser excitation was lower compared to 488 nm laser excitation (Fig. 4a, b). Quantitative analysis of the rate of cells exhibiting spiking showed that both sensors can detect calcium spikes in individual cells with similar efficiency (Fig. 4c). In the case of Matryosh-CaMP6s, the 440 nm laser excitation allowed for simultaneous

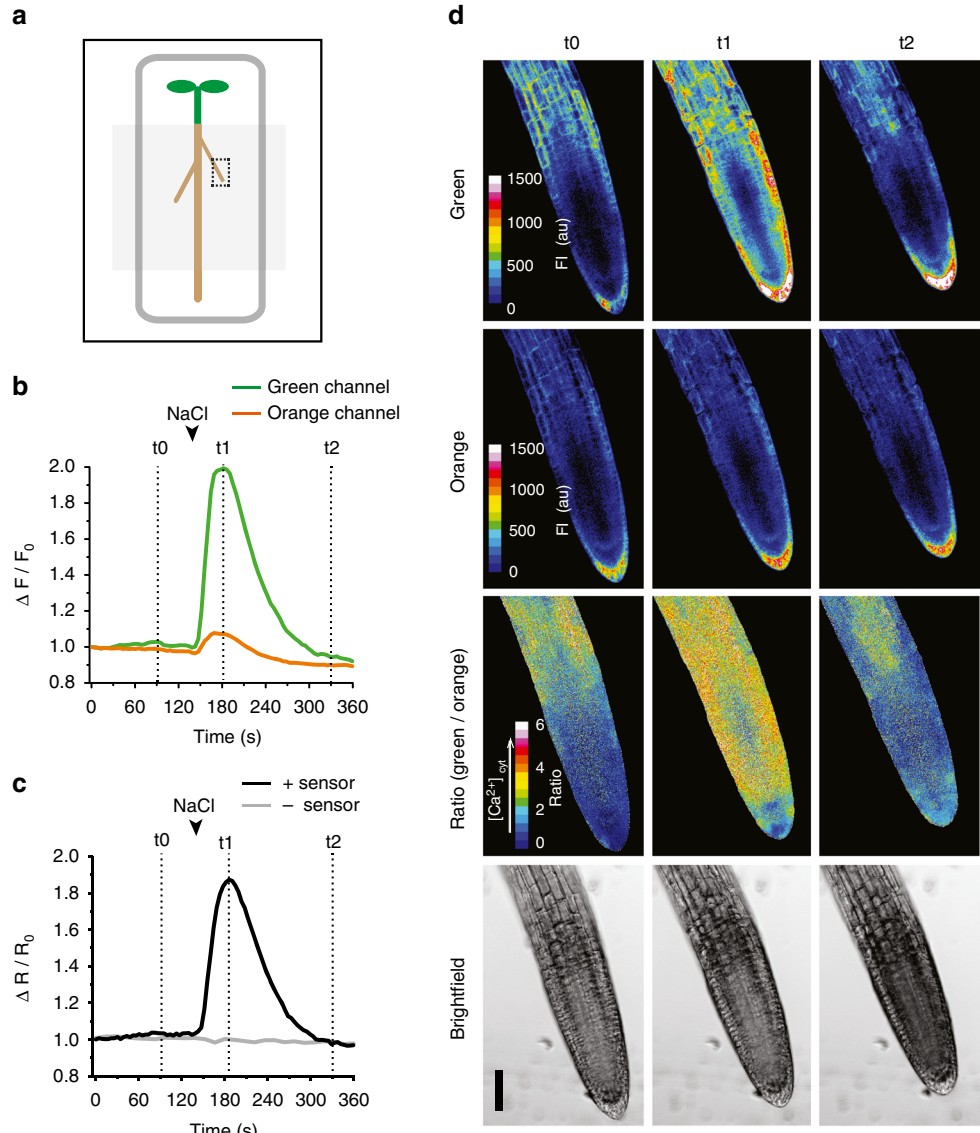

**Fig. 3** *Arabidopsis* seedlings expressing MatryoshCaMP6s report root cytosolic calcium elevation in response to salt shock. **a** Cartoon diagram of *Arabidopsis* seedling mounted on large cover slip. Media-filled reservoir was made by applying vacuum grease (*dark gray*) and a smaller cover slip (*light gray box*). Imaged region is indicated by the dotted box. **b**, **c** Sensor response, from a single seedling treatment, recorded across the entire field of view. Arrowheads mark the time (147 s) that roots were exposed to salt shock by addition of NaCl to the growth media. Final concentration in reservoir after treatment was estimated to be 50 mM. Time points (t0–t2) indicate representative images included in figure panel. Graphs display the normalized mean FI values for green and orange channels (**b**) and the normalized ratio of green to orange mean FI values (**c**, *black line*), extracted after applying a binary mask made from the orange channel to the ratiometric data. A background control seedling without the sensor (*gray line*), was used as an autofluorescence control and no ratio change was observed with treatment. **d** Average z-stack projections of confocal images showing *Arabidopsis* lateral root before NaCl treatment (t0, 100 s), during peak response (t1, 186 s), and after signal intensities returned to baseline conditions (t2, 334 s). Sample was co-illuminated with 440 and 488 nm lasers. Emission intensities were simultaneously collected at 500–540 nm (Green Channel; *first row*) and 570–650 nm (Orange Channel; *second row*) and are shown in 16-color lookup table with FI indicated in arbitrary units. The ratio of green to orange FI is shown by 16-color lookup table (Green/Orange, third row). Bright field images for each time point are shown (single slice, bottom row). *Scale bar* indicates 50 µm. At least three biological replicates were analyzed (see Supplementary Fig. 7 for additional data). All images are supported by videos (Supplementary Movies 1–4)

recording of LSSmOrange emission, providing us with the advantage for discriminating transfected cells from non-transfected cells, since GCaMP6s is dim at resting state. Also, the LSSmOrange served as stable control throughout the MeCh treatment (Fig. 4b).

Photobleaching experiments of HEK293T cells expressing GCaMP6s or MatryoshCaMP6s revealed similar half-life ($\tau_{1/2}$) values for MatryoshCaMP6s ($\tau_{1/2} = 25.5 \pm 2.8$ s; mean $\pm$ s.e.m. $n = 5$) compared to the parent sensors GCaMP6s ($\tau_{1/2} = 22.8 \pm 1.7$ s; mean $\pm$ s.e.m. $n = 5$) at 488 nm laser excitation (Fig. 4d).

We conclude from these data that LSSmOrange insertion did not have a detectable effect on the photobleaching properties of the original GCaMP6s.

**Generation of AmTryoshka sensors for ammonium transport.** A quantitative readout is necessary when biosensor expression is driven by endogenous promoters that can influence the FI of the biosensor. For example, we recently generated AmTrac ammonium transporter activity sensors, where transport and promoter

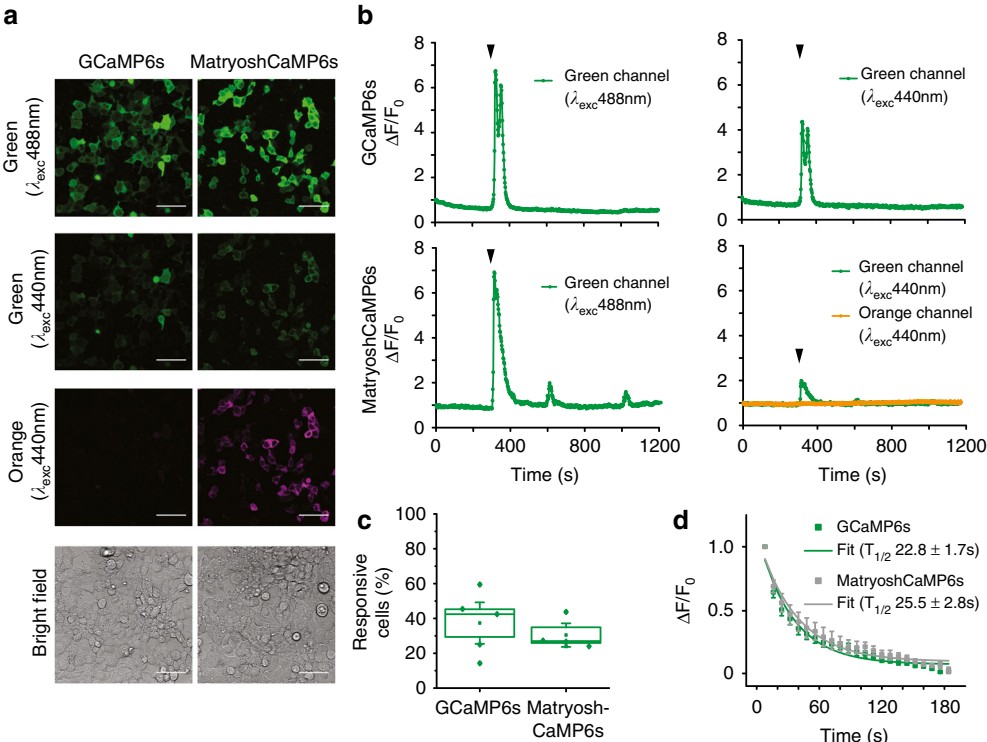

**Fig. 4** Calcium oscillations and photobleaching in HEK293T cells transfected with GCaMP6s or MatryoshCaMP6s. **a** Confocal images of HEK293T cells transfected with indicated calcium sensors and imaged at 488 or 440 laser excitation (*scale bar* = 50 μm) with detection at 500–560 nm (*green channel*) and 570–630 nm (*orange channel*). Note the cells are at resting state. **b** Calcium oscillations in HEK293T cells induced by addition of 5 μM MeCh (indicated by arrowhead) after 300 s recorded over time. Shown are exemplary graphs of the change of green and orange emission of individual cells (a minimum of three biological replicates were analyzed). Spiking was observed in the green channel, while the orange channel remained stable. Note that spiking patterns were highly variable between cells as described before[40]. **c** Quantitation of the number of cells responding to the 5 μM MeCh treatment relative to the number of cells analyzed ($n = 5$ for GCaMP6s with total of 156 cells analyzed, $n = 4$ for MatryoshCaMP6s with total of 144 cells analyzed). Note, cell response was detected under both illuminations ($\lambda_{exc}$ 488 and 440 nm, respectively). **d** Normalized decay of green emission ($\lambda_{exc}$ 488 nm) over time as a result of photobleaching. A single-exponential fit was used to calculate the half-life times ($\tau_{1/2}$) of cpEGFP in GCaMP6s and MatryoshCaMP6s (mean ± s.e. m. from biological replicates with $n = 5$)

activity can affect the FI of AmTrac. AmTrac is composed of a cpEGFP inserted into the *Arabidopsis thaliana* Ammonium Transporter 1;3 (AtAMT1;3) and reports ammonium transport in vivo by a 40% reduction of FI[25]. However, as AMT expression in plants is nitrogen-dependent[41, 42], promoter activity can affect the FI readout of AmTrac, which limits AmTrac use in planta. To address this drawback, we explored different ratiometric design approaches, including the generation of dual-emission deAm-Tracs[13] and the fusion of mCherry to the N and C terminus of AmTrac as a second, red-shifted FP. We also tested LSSmOrange as N-terminal tag of AmTrac (Supplementary Fig. 8a). Unfortunately, all resulting fusion proteins showed impaired transport activity, as evidenced by the lack of complementation in the growth assay (Supplementary Fig. 8b). Spectral analysis of yeast cultures expressing the fusion constructs demonstrated a loss of green and red or orange FI for the N-terminally labeled constructs. AmTrac with a C-terminally fused mCherry confirmed the presence of both FPs (Supplementary Fig. 8c). To restore transport activity, a suppressor screen was performed[25]. However, all the suppressing colonies that grew on ammonium showed mutations in the C terminus of AmTrac that introduced a STOP codon, resulting in the removal of mCherry and an indication that also the C-terminal mCherry was not tolerated.

The Matryoshka approach provides an alternative design for ratiometric sensors with large dynamic range and proves particularly advantageous when terminal fusion of FPs is not tolerated. Here cpEGFP in AmTrac was replaced with sfGO-

Matryoshka or sfGO-Matryosh-T78H, while retaining the flanking residue combinations that had been optimized for generating AmTrac-LS and -GS[25] (Fig. 5a). The resulting sensor series was termed AmTryoshka1;3. In parallel, we tested the properties of cpsfGFP relative to cpEGFP in AmTrac and found that the set of cpsfGFP-based AmTrac versions, named sfAmTrac, yielded improved brightness of the sensors (Supplementary Note 3).

Yeast transformed with the AmTryoshka1;3 variants showed dual-emission maxima with bright green ($\lambda_{em} \sim 510$ nm) and orange fluorescence ($\lambda_{em} \sim 570$ nm) when excited at $\lambda_{exc}$ 440 nm and a single green emission maximum ($\lambda_{em} \sim 510$ nm) at $\lambda_{exc}$ 480 nm excitation (Supplementary Fig. 9). However, we did not detect FI changes in response to ammonium addition. Insertion of the reference FP, LSSmOrange, seemed to impair transport activity of AmTryoshka1;3-GS, as confirmed by the lack of growth complementation (Fig. 5b, middle panel, second last row). To restore activity, a suppressor screen similar to previous experiments was performed[25]. Two mutations, F138I and L255I that allowed for growth on media containing low ammonium levels were identified (Fig. 5b, middle panel; mutations were numbered according to AtAMT1;3 sequence). Analysis of the crystal structure of the archaeal homolog AfAMT1 (PDB: 2B2F) indicate that both residues may line the pore (Supplementary Fig. 10). While no functional description of L255 is available, F138 had been proposed to form part of the external gate[43]. Thus, the LSSmOrange insertion may have negatively affected the gate, an effect compensated for by the F138I mutation. However, we

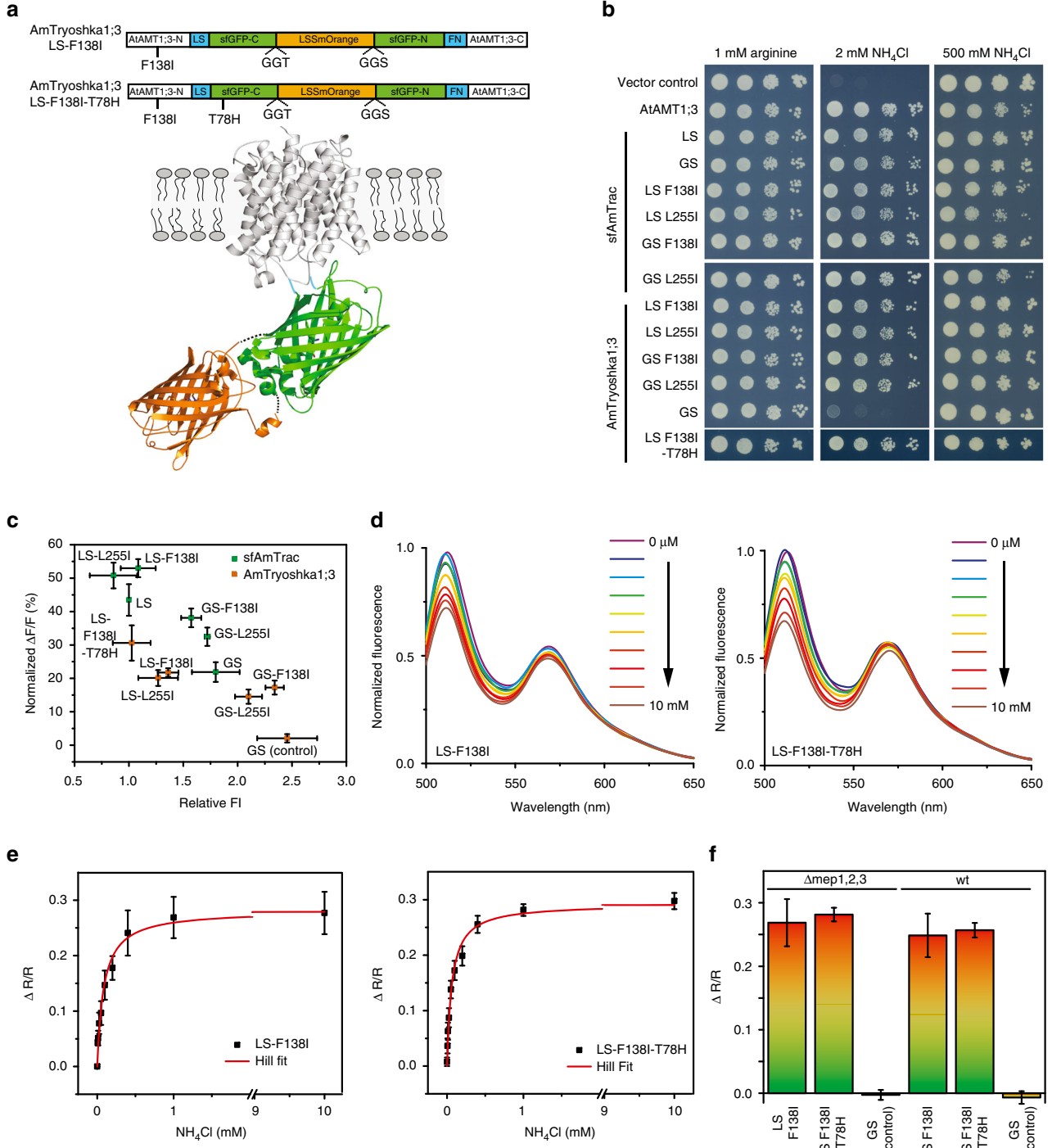

**Fig. 5** AmTryoshka characterization in yeast cell cultures. **a** Schematic representation of AmTryoshka1;3, with LSSmOrange nested inside the native C- and N-termini of sfGFP and then sandwiched between the native N and C termini of AtAMT1;3. The sensors contained the F138I suppressor mutation in the N-terminal portion of the AtAMT1;3 domain and the LS/FN residues flanking the sfGFP. AmTryoshka1;3-LS-F138I-T78H additionally contains the T78H mutation in the sfGFP domain. Cartoon illustrates only one subunit of the AtAMT1;3 trimer. **b** Complementation of the yeast $\Delta mep1,2,3$ mutant transformed with indicated sensors grown on solid media with indicated N sources. Arginine served as growth control. The columns of plated cells represent 1:10 dilutions (a minimum of three biological replicates were analyzed). **c** Relative FI (normalized to sfAmTrac-LS = 1) and fluorescence change in the green channel ($\lambda_{exc}$ 440 nm) after addition of 1 mM $NH_4Cl$ (mean ± s.d. from biological replicates with $n = 4$ for AmTryoshka1;3-LS-F138I, LS-L255I, GS-F138I, -GS-L255I, $n = 3$ for rest of constructs). **d** Steady-state emission spectra of AmTryoshka1;3-LS-F138I and -T78H with $\lambda_{exc}$ 440 nm and treatment with $NH_4Cl$ at the different concentrations (0, 6.25, 12.5, 25, 50, 100, 200, 400 μM, 1 mM or 10 mM). Spectra were normalized to maximal intensity. (**e**) Titration of $\Delta R/R$ ($R = FI_{510nm}/FI_{570nm}$) of AmTryoshka1;3-LS-F138I and -T78H (*black squares*) and Hill fit (*red line*). Data were corrected for fluorescence bleed-through (bleed-through factor 0.08) and normalized to water-treated controls (mean ± s.e.m. from biological replicates with $n = 4$ for *LS-F138I*, $n = 6$ for *LS-F138I-T78H*). **f** Fluorescence response ($\Delta R/R$) of $\Delta mep1,2,3$ or wild type (wt) transformed with AmTryoshka1;3-LS-F138I, -T78H or the non-responsive control AmTryoshka1;3-GS (mean ± s.e.m. from biological replicates with $n = 4$ for LS-F138I, $n = 5$ for LS-F138I-T78H, $n = 3$ for GS of $\Delta mep1,2,3$ transformation, $n = 5$ for LS-F138I, $n = 4$ for LS-F138I-T78H, $n = 4$ for GS of wild-type transformation)

did not detect any changes in the plasma membrane localization in yeast compared to non-mutated sfAmTrac-LS (Supplementary Fig. 11a).

Steady-state analysis of ammonium titrations in yeast expressing either AmTryoshka1;3-F138I or -L255I revealed a reduction in FI in the green channel by 15–17% for the GS-flank variant and 20–22% for the LS-flank variant (Fig. 5c, d; Supplementary Fig. 11b). The AmTryoshka1;3-LS-F138I-T78H mutant showed an ammonium-induced reduction in green FI of 30%, thus providing the highest dynamic range (Fig. 5c, d). Quantitative analysis of the ratio change ($\Delta R/R$) in response to ammonium titration demonstrated unaltered ammonium affinities ($K_{0.5}$) compared to AmTrac, indicating that the dual-FP construct can ratiometrically report ammonium transport[25] (Fig. 5e; Supplementary Fig. 11c; Supplementary Table 3).

To exclude environmental effects, such as accumulation of intracellular ammonium, which could affect the cytosolic pH, wild-type yeast with endogenous MEP ammonium transport activity was transformed with AmTryoshka1;3-LS-F138I, AmTryoshka1;3-LS-F138I-T78H or the non-responsive control AmTryoshka1;3-GS. In the presence of endogenous MEP ammonium transporter, cytosolic ammonium concentrations are expected to increase independently of the sensors. Sensor responses were comparable between the wild-type strain and the $\Delta mep1,2,3$ mutant (Fig. 5f), indicating that intracellular ammonium levels did not affect the sensor. Hence, using Matryoshka technology, we engineered ratiometric ammonium transport sensors that report substrate concentration-dependent ammonium transport activity. The sensor with the highest dynamic range was AmTryoshka1;3-LS-F138I-T78H, with an approximate 30% FI decrease in response to ammonium transport.

Despite the insertion of the sfGO-Matryoshka cassette and the introduction of suppressor mutations, the transport activity of AmTryoshka1;3-LS-F138I-T78H as measured by two-electrode voltage clamping in oocytes was not affected when compared to the parental AtAMT1;3 transporter or AmTrac (Supplementary Fig. 12)[25].

## Discussion

Here we developed a platform for ratiometric biosensor design. We engineered two cassettes, eGO-Matryoshka and sfGO-Matryoshka, carrying dual-emission FP combinations of a nested LSSmOrange in either cpEGFP or its superfolder variant cpsfGFP, respectively. The LSS of LSSmOrange[28] enabled single wavelength excitation ($\lambda_{exc}$ 440 nm) of both green and orange fluorescence. Excitation at $\lambda_{exc}$ 480–488 nm yielded green fluorescence only, with minimal cross-excitation of LSSmOrange. Nesting of LSSmOrange was well tolerated in both cpFPs, as evidenced by the lack of detectable differences in the fluorescence properties, including chromophore maturation efficiencies of eGO-Matryoshka or sfGO-Matryoshka compared to the free FPs (cpEGFP, cpsfGFP or LSSmOrange).

As test case, we utilized the Matryoshka technology and converted the highly sensitive calcium sensor GCaMP6s into MatryoshCaMP6s in a single step. Three MatryoshCaMP6s versions (MatryoshCaMP6s, sfMatryoshCaMP6s and sfMatryoshCaMP6s-T78H) were generated, which differed in their sensitivities and affinities toward calcium ($K_d$ 200–500 nM). In vitro, the MatryoshCaMP6s iterations demonstrated similar spectral and kinetic properties as the respective parents. Interestingly, the different cpFPs also affected the dynamic range of the sensor iterations.

Our mathematical model (Supplementary Note 2) indicates that three mechanisms appear to differentially affect the dynamic range of the individual calcium sensors: (i) $pK_a$ differential (ii)

relative quantum yield and (iii) internal acid-base equilibrium between the apo and saturated species. While GCaMP6s is modeled to utilize all three mechanisms, sfGCaMP6s seems to derive most of its response from their greater $pK_a$ differences between apo and saturated proteins. Since the theoretical dynamic range is the product of the three factors, even small differences can substantially affect sensor responses (Supplementary Note 2). Analysis of crystal structure and systematic mutagenesis followed by spectral characterization and model fitting will help to parse the molecular connections to the three principal determinants of the dynamic range and may thus aid in future optimization of such sensors. Single-FP biosensors are often limited in their response accuracy due to pH sensitivity. The newly identified internal equilibrium factor is expected to be largely independent of cytosolic pH over normal physiological ranges. Thus, we predict that biosensors which maximize the quantum yield difference and the internal equilibrium factor can reach large dynamic ranges while also retaining high signal fidelity even in the presence of cellular pH fluctuations.

We demonstrated that MatryoshCaMP6s can be used to monitor biological processes in intact organisms, such as *Arabidopsis* seedlings. MatryoshCaMP6s offers several advantages over previously developed tools in this context. A co-localized reference FP is particularly important for long-term acquisition where quantitative data is desired. Because GCaMP sensors are dim at resting cytosolic calcium levels, we expect that MatryoshCaMP6s calcium sensors will be more suitable for verifying cell-type specific expression. Also for this reason, signal from the reference FP enables more confident interpretation of negative results (lack of intensity changes) in the reporting FP.

We verified that the Matryoshka technology is suitable for constructing ratiometric membrane transporter biosensors. It is worth noting that AtAMT1;3 is extremely sensitive toward modifications[25]. Therefore, it is not surprising that insertion of a second FP affected the transporter function. The initial impairment of AmTryoshka1;3 transport activity, and thus lack of FI signal change, triggered by the LSSmOrange insertion, was overcome by the individual suppressor mutations F138I and L255I, which restored transport function and sensor response.

Additionally, our results provide insights into the role of histidine 78 in cpFP. For both calcium and ammonium transport activity sensors, the T78H mutation in cpsfGFP yielded a larger dynamic range relative to sensors with the T78. Histidine at position 78 was postulated to be involved in excluding solvent from the FP-chromophore[34]. However, in previous work, the effects of T78H could not explicitly be assigned to histidine 78 only, since GCaMP6s contained additional mutations compared to other GCaMP6 versions[6]. Here we could show that histidine 78 of cpEGFP and cpsfGFP results in a gain of sensor sensitivity. Both cpsfGFP-T78H and sfGO-Matryoshka-T78H, containing the T78H mutation in the cpsfGFP-domain, provide improved FP components for more sensitive sensor designs in future permutations.

The multiple iterations of MatryoshCaMP6s and AmTryoshka1;3 now contain an internal reference and are highly effective in two operation modes: (i) at excitation $\lambda_{exc} \sim 440$ nm, the ratio of green-orange emission offers signal normalization, and thus improved quantitation, and (ii) excitation at $\lambda_{exc} \sim 488$ nm allows for maximal dynamic range and sensitivity, equivalent to the parent sensors.

The modular design of Matryoshka, with the reference FP nested within the center of the fused protein, offers several advantages over biosensors with co-expressed or terminally fused reference FPs: (i) as opposed to proteins expressed from bicistronic plasmids, Matryoshka is a single protein where all protein moieties will be produced and degraded simultaneously, allowing

for accurate quantitation if appropriate analyte calibration procedures are available. (ii) it requires only one insertion site in the binding protein, as fusion of multiple FPs is often not tolerated, especially for membrane transporters. (iii) the single cassette facilitates construction of insertion libraries from which to rapidly obtain effective biosensors[44], and (iv) it reduces artifacts in the ratiometric readout that may occur from proteolysis of terminal FP domains. Matryoshka thus represents a versatile tool for generating ratiometric biosensors in a single cloning step.

For reliably accurate analyte quantitation, it should be considered that individual FPs differ in their properties, such as chromophore maturation, demonstrated by our maturation analysis and photobleaching characteristics, and nesting of FPs will maintain these limitations. Future generations of novel FPs with more advantageous properties will substantially improve Matryoshka technology. To eliminate the potential phototoxic effects of $\lambda_{exc}$ 440 nm of the current Matryoshka sensors, further efforts will be invested to red-shift the excitation properties. Alternatively, two-photon excitation should be investigated. Additionally, the Matryoshka concept will be used to improve the fluorescence properties of sensors for other small molecules such as sugars, amino acids, and hormones[22, 23, 45, 46].

## Methods

**DNA constructs.** For in vitro characterization of the single FPs, pET15b-cpsfGFP-LS-FN was generated by modifying the circular permutation breakpoint of the cpsfGFP sequence in the bacterial expression vector pET15b via site-directed mutagenesis. Primer pair LS-cpsfGFP_FW/LS-cpsfGFP_RV was used to replace the NSH sequence with LS, and primer pair cpsfGFP-FN_FW/cpsfGFP-FN_RV replaced the F with FN (see Supplementary Table 4 for primers). Similarly, pET15b-cpsfGFP-LE-LP was generated using primer pairs LE-cpsfGFP_FW/LE-cpsfGFP_RV and cpsfGFP-LP_FW/cpsfGFP-LP_RV (Supplementary Table 4). pET15b-eGFPcp-LE-LP and pET15b-eGO-Matryoshka-LE-LP were created through PCR amplification of the FP sequences of pRSETa-GCaMP6s and pRSETa-MatryoshCaMP6s with the primer pair pET-EGFPcp InF_FW/pET-EGFPcp_InF_RV (Supplementary Table 4). pET15b-cpsfGFP-LE-LP was then digested with XhoI (New England Biolabs, Ipswich, MA) and an In-Fusion® (Clontech, Mountain View, USA) reaction was performed to recombine the fragments. pET15b-LSSmOrange was generated by an initial PCR amplification of the LSSmOrange sequence using the primers LSSmOr-pET15b_InF_1st_FW, containing a HIS tag overhang, and LSSmOr-pET15b_InF_1st_RV, adding a stop codon (Supplementary Table 4). A second round of PCR amplification with primers LSSmOr-pET15b_InF_2nd_FW and LSSmOr-pET15b_InF_RV was performed to add overlaps for subsequent In-Fusion HD cloning (Clontech; Supplementary Table 4). pET-15b-cpsfGFP was digested with XhoI and NcoI-HF (NEB) to remove the cpsfGFP. In-Fusion cloning was performed per Clontech's protocol to recombine the purified fragments.

sfGO-Matryoshka was created by digesting the pET-15b-cpsfGFP plasmids and the pDRF'-AmTryoshka1;3-GS construct with AgeI-HF and DraIII-HF (NEB), followed by gel-purification with a commercial kit (Machery-Nagel, Düren, Germany) and ligation by T4 DNA ligase (Thermo Scientific), subsequently inserting the LSSmOrange into the center of the cpsfGFP-GGT-GGS flexible linker.

For the generation of sfAmTrac-LS and GS, overlap-PCR was employed to exchange cpEGFP for cpsfGFP. Briefly, three DNA fragments were generated, the N-terminal AtAMT1;3 fragment (amino acids 1-233), the C-terminal AtAMT1;3 fragment (amino acids 234–498) and the cpsfGFP fragment. cpsfGFP was amplified from the pET15b-cpsfGFP with the forward primer AmLS_sfGFPcp_FW or AmGS_sfGFPcp_FW, including the coding sequence for the LS flank or GS flank, respectively, to replace the NSH flank on the N-terminus of the cpsfGFP, and the reverse primer coding for FN AmFN_sfGFPcp_RV to replace the F flank on the C-terminus of cpsfGFP sequence (Supplementary Table 4). Thus, the cpsfGFP contains the equivalent breakpoint in the sfAmTracs as the original AmTracs[25]. The fragments were combined into the pDONR-221 vector via Gateway BP-reaction and then moved into pDRF'-GW via Gateway LR reaction (Invitrogen Life Technology, Paisley, UK).

The AmTryoshka1;3-GS sequence was synthesized using GeneScript and introduced into pDRF'-GW vector via Gateway reaction (Invitrogen Life Technology). pDRF'-AmTryoshka1;3-GS served as the basis for the generation of the different AmTryoshka variants.

AmTryoshka1;3-LS-F138I and –L255I as well as sfAmTrac-GS-F138I and -L255I and sfAmTrac-LS-F138I and -L255I were generated via site-directed mutagenesis performed according to the guidelines of the QuikChange II XL Site-Directed Mutagenesis Kit (Stratagene, Agilent Technologies, Santa Clara, USA). Primers sfLS-LSSmO_FW and sfLS-LSSmO_RV exchanged the GS sequence for LS; primers sfAmTrac-F138I_FW and sfAmTrac-F138I_RV introduced the F138I

mutation; and primers sfAmTrac-L255I_FW and sfAmTrac-L255I_RV introduced the L255I mutation (Supplementary Table 4).

Calcium sensor variants were cloned by digesting the full calcium sensor sequence out of pGP-CMV-GCaMP6s[6] (Addgene plasmid #40753) with MfeI and NheI-HF and ligating into the bacterial expression vector pRSETa linearized with NheI-HF and EcoRI-HF. pRSETa-MatryoshCaMP6s was produced by inserting a PCR-amplified LSSmOrange into the middle of the GGT-GGS flexible linker of the KpnI-digested pRSETa-GCaMP6s construct via In-Fusion (GCaMP6s-EGFPcp-LSSmO-InF_FW and GCaMP6s-EGFPcp-LSSmO-InF_RV). pRSETa-sfGCaMP6s and pRSETa-sfMatryoshCaMP6s were assembled by substituting the cpEGFP of GCaMP6s with either a cpsfGFP or a sfGO-Matryoshka. The full-length sequences of cpsfGFP and sfGO-Matryoshka were individually PCR amplified with overlaps containing 9 bp of the 3′ end of the M13 peptide and XhoI restriction site/LE amino acid residues at the 5′-terminal as well as the C-terminal LP amino acid residues and 14 bp of the 5′ end of the calmodulin protein at the 3′-end (sfGFPcp-XhoI-M13-InF_FW and sfGFPcp-LP-CaM_RV). Another PCR fragment was generated with the full GCaMP6s calmodulin protein containing 21 bps of overlap with the cpsfGFP 3′-end (using CaM-LP-sfGFPcp_FW and CaM-pRSET-HindIII-InF_RV.). The two fragments were then ligated via a two-step PCR protocol. The resulting PCR product was recombined by In-Fusion (Clontech) into pRSETa-GCaMP6s that had been digested with XhoI and HindIII-HF.

The T78H mutation was introduced into constructs containing a cpsfGFP via site-directed mutagenesis.

For replacement of the nested reference FP LSSmOrange with cyOFP1, Primestar Max DNA Polymerase (Clontech) was used to amplify linear fragments that were recombined by In-Fusion HD enzyme kit (Clontech). Each pRSET-MatryoshCaMP6s vector backbone was amplified using outward-facing primers (cpGFP-IF1 and cpGFP-IR1 for cpEGFP construct; sfcpGFP-IF1 and sfcpGFP-IR1 for cpsfGFP constructs; Supplementary Table 4). The coding sequence of cyOFP1 was amplified from the pNCS-cyOFP1 vector (a kind gift from Michael Lin, Stanford University) using primer pairs with 18-nucleotide 5′ extensions homologous to vector backbone fragments (cyOFP1-F1h1 and cyOFP-R1h1 for cpGFP constructs; cyOFP1-F1h2 and cyOFP-R1h2 for sfcpGFP constructs; Supplementary Table 4). Plasmid templates were digested by DpnI FastDigest (Thermo Fisher) before In-Fusion cloning.

For expression in plants, MatryoshCaMP6s was inserted into the Arabidopsis UBQ10 driven binary vector pGPTVII-Bar-U (a kind gift from Melanie Krebs, University of Heidelberg, Heidelberg[47]). MatryoshCaMP6s was PCR amplified from pRSETa MatyroshCaMP6s and inserted into the SpeI-HF/XmaI (New England Biolabs) digested pGPTVII-Bar-U backbone via InFusion (Clontech).

**Protein expression and purification.** BL21 (DE3) cells were transformed with the FP constructs in the bacterial expression vector pET-15b and GCaMP6s variants in pRSETa. Single colonies were grown in Luria broth containing 50 µg/ml carbenicillin at 20 °C and were incubated in the dark for 48 h. Cells were harvested by centrifugation and frozen at −20 °C overnight. Pellets were resuspended in 5 ml buffer (20 mM Tris-HCl pH 8), disrupted via sonication, and centrifuged for 1 h at 4100 rpm and 4 °C to remove cellular debris. The lysate was filtered through 0.45 µ and applied to 2 mL Novagen HIS-Bind Resin (cat. #69670 EMD Millipore) charged with 50 mM NiCl₂ in Bio-Rad gravity columns (cat. #731-1550 Bio-Rad). Columns were washed twice with buffer (20 mM Tris-HCl pH 8) and eluted in 1.5–2 ml 200 mM imidazol in 20 mM Tris-HCl pH 8. Purified protein was then allowed to mature overnight at 4 °C before performing measurements. Eluted protein was quantified in accordance with Thermo Scientific's Coomassie (Bradford) Protein Assay kit (Thermo Scientific, Waltham, MA, USA).

**UV/Vis absorbance and mass spectrometry of FPs and FP-fusion.** Individual FPs (cpEGFP-LE-LP, cpsfGFP-LE-LP and LSSmOrange) and FP-fusions (eGO-Matryoshka-LE-LP and sfGO-Matryoshka-LE-LP) were further purified by anion exchange chromatography (HiTrap Q HP column, GE Healthcare). All proteins showed a single peak in the chromatogram, apart from LSSmOrange which showed two peaks in the chromatogram.

Absorbance spectra of all fractions at a concentration of ~ 10–0.5 µM were collected with a PerkinElmer Lambda 25 UV–vis spectrometer using 1 ml silica cuvettes and scanning the range of 250 to 550 nm with a 1-nm step size.

The protein fractions were analyzed via electrospray ionization mass spectrometry at the Mass Spectrometry facility at Stanford using a Bruker micrOTOF-Q II benchtop Q-Tof, operated in conjunction with an Agilent 1260 UPLC-UV-MS.

To determine the extinction coefficients of cpEGFP, cpsfGFP and LSSmOrange, the FPs were diluted 20-fold, denatured in 0.2 M NaOH and the absorbance was monitored.

The results represent two biological replicates.

**Fluorimetric analyses of fluorescent sensors.** All in vitro fluorimetric analyses were carried out using a fluorescence plate reader (Infinite, M1000 Pro or Safire; Tecan, Switzerland). Calcium titrations used commercial Calcium Calibration Buffer Kit #1 (Invitrogen Life Technology, Paisley, United Kingdom). The stock solutions of zero-free calcium buffer (10 mM K₂EGTA, 100 mM KCl, 30 mM MOPS pH 7.2) and 39 µM calcium buffer (10 mM Ca-EGTA, 100 mM KCl, 30 mM

MOPS pH 7.2) were mixed according to the manufacturer, yielding 11 different free calcium concentrations. 10 µl of purified protein sample was added to 90 µL of buffer zero-free calcium buffer or 39 µM calcium buffer to yield a final protein concentration of 1–1.5 µM and analyzed in 96-well, black, flat-bottom plates (Greiner Bio-One, Germany). Steady-state fluorescence spectra were recorded in bottom reading mode using 5 nm bandwidth and a gain of 90 for both excitation and emission wavelengths ($\lambda_{exc}$ = 440 or 485 nm; $\lambda_{em}$ = 525 or 570 nm). Spectra were background-subtracted using a buffer control and values of emission maxima were extracted for dynamic range calculations and plots ($\Delta F/F_0$; $\Delta R/R_0$). Throughout the measurements, the temperatures ranged between 25–35 °C, and free calcium concentrations were adjusted accordingly[48]. Correction for fluorescence bleed-through, with a calculated bleed-through factor of ~ 0.10 of green fluorescence in the orange emission channel, was performed prior to fitting the titration kinetics by a sigmoidal dose response function.

pH titrations were performed for purified calcium sensors and FPs. Calcium sensor protein was added to a final concentration of 1 µM (fluorescence) and 4–6 µM (absorbance) in pH buffers containing 50 mM citrate, 50 mM Tris, 50 mM glycine, 100 mM NaCl, and either 5 mM $CaCl_2$ or 5 mM EGTA. pH buffers with $CaCl_2$ were adjusted to 11 different pH values from 4.5–10. pH buffers containing EGTA were adjusted to 11 different pH values from 5–10.5. pH titrations of FPs were carried out by diluting purified protein to a final concentration of 1 µM into the pH buffers containing 10 mM citric acid (pH 2.5) premixed with 100 mM sodium phosphate dibasic (pH 9) to yield 11 solutions of pH buffers ranging from pH 4–9. Glycine-NaOH pH buffer was used to attain pH values 9.5–10.5. Analysis was performed in 96-well, black, flat-bottom, half-area plates (Greiner Bio-One, Germany). Steady-state fluorescence spectra were recorded in bottom reading mode using 5 nm bandwidth and a gain of 90 for both excitation and emission wavelengths ($\lambda_{exc}$ = 440 or 485 nm; $\lambda_{em}$ = 525 or 570 nm). Absorbance scans were obtained over the range of 270 to 550 nm with a 5-nm step size. Spectra were background-subtracted using a buffer control. Values of emission maxima ($\lambda_{exc}$ = 440/485 nm; $\lambda_{em}$ = 510/515) were extracted to plot the pH titrations. A Boltzman fit was used for the $pK_a$ calculations.

Steady-state emission measurements of liquid yeast cultures expressing AmTrac-LS-mCherry were acquired using a Fluoromax-P fluorescence spectrometer (Horiba Jobin Yvon, Kyoto, Japan) and 3.5 mL silica cuvettes (Hellma Analytics, Mullheim, Germany). For the cpEGFP fluorescence, the excitation and emission spectra were recorded at $\lambda_{em}$ = 514 nm and $\lambda_{exc}$ = 485 nm, respectively. mCherry fluorescence was recorded at $\lambda_{em}$ = 610 nm and $\lambda_{exc}$ = 585 nm, respectively. A step size of 1 nm was chosen and five repeats were taken for averaging. Untransformed yeast cells served as blank.

All ammonium titrations were performed using a fluorescence plate reader (Safire; Tecan, Männedorf, Switzerland). Washed yeast cells (200 µl) expressing the sfAmTrac and AmTryoshka1;3 variants were loaded into black, 96-well microplates with clear bottom (Greiner bio-one, Germany). For the titrations, 50 µl of $NH_4Cl$ were added to the cells to final ammonium concentrations of 0, 6.25, 12.5, 25, 50, 100, 200, 400 µM, 1 or 10 mM (water added for the zero value). Cells were incubated for 8 min to saturate the response. Steady-state fluorescence was recorded in bottom reading mode using 7.5 nm bandwidth and a gain of 100. The fluorescence emission spectra ($\lambda_{exc}$ = 440 or 480 nm) and single point values ($\lambda_{exc}$ = 440 or 485 nm; $\lambda_{em}$ = 510 or 570 nm) were background subtracted using yeast cells expressing a non-florescent vector control. Correction for fluorescence bleed-through, with a calculated bleed-through factor of ~ 0.08 for green fluorescence in the orange emission channel, was performed prior to $\Delta R/R$ calculations (R = $FI_{510nm}/FI_{570nm}$). The affinity constants ($K_{0.5}$) were determined using a fit to the Hill equation.

All graphs and spectral analyses were performed using OriginPro 2015 software (OriginLab, Northampton, MA, USA).

**Plant growth and transformation.** Transgenic *Arabidopsis* lines containing MatryoshCaMP6s driven by the AtUBQ10 promoter were generated by *Agrobacterium*-mediated floral dip into the Col0 rdr6-11 silencing mutant[49]. T1 seeds were selected for successful transformants on half-strength MS medium with vitamins (PhytoTechnology Laboratories, Shawnee Mission, KS; cat. no. M519) containing 1% agar (Sigma, cat. no. 9002-18-0,) 1% sucrose, 2.3 mM MES pH 5.7 and 50 µg/ml BASTA.

Progeny of the selected T1 transformants were germinated for imaging on half-strength MS medium (PhytoTechnology Laboratories, cat. no. M524) containing 1% agar (Sigma) and 2.3 mM MES pH 5.7. Seedlings were grown vertically under long-day light conditions at a light intensity of about 100 µmol/m² sec for 7–9 days before imaging. To confirm fluorescent signal, seedlings were preliminarily screened for GFP fluorescence using a Nikon SMZ18.

**Plant imaging conditions.** Images were collected on a Leica TCS SP8 equipped with resonant scanner and white light laser (WLL). A HC PL APO 20 × /0.70 N.A. multi-immersion objective was used with glycerol. Scan speed was 8000 Hz (resonant mode), and line averaging was set to 8 or 16. Samples were co-excited with the 440 nm pulsed laser and the 488 nm laser line of the WLL. Fluorescence images were captured simultaneously in two windows using HyD SMD detectors; gain was set to 90 for each. Signal from cpEGFP was collected from 500–540 nm.

Signal from LSSmOrange was collected from 570–650 nm. Transmitted light images were collected with a PMT.

Seedlings were prepared for imaging by gentle transfer to cover slips and stabilized with surgical glue (spray PDMS), as described previously[50]. The reservoir for liquid media was made using vacuum grease and filled with half-strength MS medium without sucrose. An additional cover slip was placed over the seedling to hold it in place and facilitate movement of treatment to the root. Salt shock was applied by addition of half-strength MS containing NaCl to the liquid/cover glass interface. Final concentrations of NaCl in the seedling reservoirs were ~10–50 mM, depending of the volume of media required for sample preparation.

At least three biological replicates were analyzed. Images of treatments were excluded from data analysis if extreme shifting of the root occurred upon application of NaCl or the application of treatment did not reach the media within the reservoir.

Average intensity z-stack projections were generated using FIJI (http://fiji.sc). The average ratio of cpEGFP / LSSmOrange signal intensities were calculated using FIJI, and ratio values are portrayed in pseudocolor (16-color lookup table). A binary mask was made using the LSSmOrange channel and applied to the ratio values to remove noise from background signal (Supplementary Movie 5). The average pixel ratio value per time point was extracted from the masked dataset, normalized with the initial ratio set to 1, and plotted using OriginPro 2015 (OriginLab).

**HEK293T cell assay and imaging conditions.** Transfection of HEK293T human kidney epithelial cells (ATCC, cat. no. CRL-11268) was performed as described previously[51]. Briefly, cultured cells were seeded in a collagen-coated 8-well chamber (NUNC Lab-Tek II 155409) 24 h prior transfection. For the transfection, the calcium sensor plasmids in the pcDNA3.2-DEST (Thermo Fisher Scientific, cat. no. 12489019) vector backbone were mixed with the transfections reagents according to the manufacturer guidelines (Invitrogen). The cells were cultured in 250 µL Opti-MEM media (Gibco, cat. no. 31985-062) overnight at 37 °C and 5% $CO_2$. Prior imaging, the media was replaced for 225 µL Hank's buffered saline solution (HBSS) with $CaCl_2$ and $MgCl_2$ (Gibco, cat. no. 14025-092).

Fluorescence images and time-series were acquired with a 40 x oil objective (NA 1.25) on a White Light Laser Confocal Microscope Leica TCS SP8 X using the 488 nm laser line. In addition, the 440 nm pulsed laser was used for parallel excitation and detection of the cpEGFP or cpsfGFP and LSSmOrange. The HyD detector range was set to 500–560 nm for cpEGFP or cpsfGFP and 570–630 nm for LSSmOrange detection, respectively.

For calcium oscillations, a time course of a single plane was recorded for a duration of 20 min with the manual addition of 25 µL 50 µM acetyl-β-Methacholine chloride (Sigma cat. No. 2251–25G) at time point 150 (5 min). Images were recorded using the sequential scan mode at a time interval of 2 s per scan. Evaluation of calcium oscillations was performed using the FIJI software (http://fiji.sc). The ROI manager tool was used to quantify the average pixel intensity of individual cells and the results were plotted over time.

For photobleaching experiments, a time course of z-stacks (30 slices at 37 µm) was recorded for a duration of 3 min. The 488 nm laser excitation was set to full power. Images were recorded using the sequential scan mode at a time interval of 8 s per z-stack scan. Evaluation of photobleaching was performed using FIJI. The z-stacks were transformed to an average intensity projection and a mask was created to identify transformed cells. The ROI manager tool was used to quantify the average pixel intensity of individual cells and the results were plotted over time and averaged. Random calcium spikes were observed during the bleaching experiments and such cells traces were removed before averaging. Results of a minimum of three transformations were analyzed and graphs were plotted using OriginPro 2015 software.

**Yeast transformation and culture.** The in vivo measurements employed the yeast strain 31019b [*mep1Δ mep2Δ::LEU2 mep3Δ::KanMX2 ura3*][52], which lacks all endogenous MEP ammonium transporters[30, 53]. Briefly, yeast transformation was performed using the lithium acetate protocol[54]. Transformants were plated on solid YNB (minimal yeast medium without amino acids/ammonium sulfate; Difco BD, Franklin Lakes, NJ) supplemented with 3% glucose and 1 mM arginine. Single colonies were selected and inoculated in 5 ml liquid YNB supplemented with 3% glucose and 0.1% proline under agitation (230 rpm) at 30 °C until $OD_{600nm}$ 0.5–0.9.

AmTryoshka1;3-GS, which did not show a response upon ammonium treatment, was subjected to a suppressor screen as previously described[25]. Briefly, liquid cultures of yeast cells expressing AmTryoshka1;3-GS were washed twice with sterile water. The final resuspension volume was 5 mL and 500 µL were streaked on five plates with a diameter of 150 mm (VWR, Radnor, PA, USA) of solid YNB medium (buffered with 50 mM MES/Tris, pH 5.2, supplemented with 3% glucose and 1 mM $NH_4Cl$). The plates were incubated at 30 °C; single colonies were identified after 7 days. Yeast plasmid DNA was isolated and sequenced, revealing the mutations F138I and L255I. AmTryoshka1;3-GS with a mutation was called AmTryoshka1;3-GS-F138I or –L255I, respectively.

For complementation assays, liquid cultures were diluted $10^{-1}$, $10^{-2}$, $10^{-3}$ and $10^{-4}$ in water, from which 5 µl of each dilution was spotted onto solid YNB medium (buffered with 50 mM MES/Tris, pH 5.2 supplemented with 3% glucose). Either $NH_4Cl$ (2 mM; 500 mM) or 1 mM arginine was added as sole nitrogen

source. After 3 days of incubation at 30 °C, cell growth was documented by scanning the plates at 300 d.p.i. in grayscale mode.

For fluorescence measurements, liquid yeast cultures were washed twice in 50 mM MES pH 6.0, and resuspended to $OD_{600nm} \sim 0.5$ in MES pH 6.0, supplemented with 5% glycerol to delay cell sedimentation[53]. All graphs and were plotted using OriginPro 2015 software.

**Yeast imaging conditions**. Confocal z-sections of yeast cells expressing the sfAmTrac and AmTryoshka1;3 sensor variants were acquired with a 63 x oil objective (NA 1.40) on a White Light Laser Confocal Microscope Leica TCS SP8 X using the 488 nm laser line and a 440 nm pulsed laser in sequential scan mode. The HyD detector range was set to 500–560 nm for cpEGFP or cpsfGFP and 570–630 nm for LSSmOrange detection, respectively. Cells from three individual transformations were imaged. Images were analyzed using FIJI.

**Electrophysiology of AmTryoshka1;3 in Xenopus oocytes**. Two-electrode voltage clamping in oocytes was performed as described previously[25]. In brief, for in vitro transcription, pOO2-AmTryoshka1;3-LS-F138I-T78H was linearized with *Mlu*I. Capped cRNA was in vitro transcribed by SP6 RNA polymerase (mMES-SAGE mMACHINE, Ambion, Austin, TX). *Xenopus laevis* oocytes were obtained from Ecocyte Bio Science (Austin, TX). Oocytes were injected via the Roboinjector (Multi Channel Systems, Reutlingen, Germany;[55, 56] with distilled water (50 nl as control) or cRNA from AmTryoshka1;3-LS-F138I-T78H (50 ng in 50 nl). Cells were kept at 16 °C 2–4 days in ND96 buffer containing 96 mM NaCl, 2 mM KCl, 1.8 mM CaCl₂, 1 mM MgCl₂, and 5 mM HEPES, pH 7.4, containing gentamycin (50 μg/μl) before recording experiments. Recordings were typically performed at day three after cRNA injection. Electrophysiological analyses of injected oocytes were performed as described previously[25, 57]. Reaction buffers used recording whole-cell currents from the injected *Xenopus* oocytes at a holding potential of −120 mV were (i) 230 mM mannitol, 0.3 mM CaCl₂, and 10 mM HEPES and (ii) 220 mM mannitol, 0.3 mM CaCl₂, and 10 mM HEPES plus indicated concentrations of NH₄⁺. Typical resting potentials were ~40 mV measured by the two-electrode voltage-clamp Roboocyte system (Multi Channel Systems)[55, 56].

**Data availability**. The plasmids generated during this study are available from Addgene. All other data that support the findings of this study are available from the corresponding author on reasonable request.

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

## Acknowledgements

We gratefully acknowledge support to W.B.F. from the National Science Foundation (MCB-1413254). L.M.O. acknowledges the support of an NSF Graduate Research Program Fellowship. We thank Prof. Steven Boxer of Stanford University for helpful advice and support through NIH Grant GM27738. We thank Theresa McLaughlin from the Stanford University Mass Spectrometry facility for the mass spectrometry analysis.

## Author contributions

C.A., L.M.O., R.D.M. and W.B.F. designed the research. C.A., J.F. and R.D.M. performed the experiments. C.-H.H. performed the oocyte experiments. C.A., J.F., L.M.O., R.D.M., T.J.K. and W.B.F. analyzed the data. C.A., L.M.O. and W.B.F. wrote the manuscript.
