## [Peer Review File · Nature Communications]

Reviewers' comments:

Reviewer #1, an expert in Ca²⁺ biosensors (Remarks to the Author):

This manuscript details the development of a new dual FP approach to the generation of fluorescent protein-based biosensors. Although there are a wide range of these kinds of sensors that have been developed, the current work makes important advances to the technology available by generating sensors with inherently large dynamic ranges of signals, with an internal reference wavelength and that are compatible with fusions to some sensor domains that are intrinsically hard to tag at the N- or C-termini. This is a clever nested cpGFP approach that is both novel, useful and potentially widely applicable. In addition, the paper uses a modeling approach to help understand the complex interactions that lead to signal changes within a sensor protein, revealing new information about the role of an internal acid-base equilibrium and helping advance the theory behind how these sensors work. The paper is well presented and makes a convincing case for the utility of the new nested fluorescent protein approach. I have a few minor suggestions that might help make the message from a well presented piece of work a little more strongly but overall, this is a comprehensive body of work that should be of great interest to the readers of Nature Communications.

It would be very useful to add empirical pH titration curves. There is a lot of pH dynamic analysis in the Appendix, but the cpGFPs are often very pH sensitive and effects of pH on each signal peak intensities will be important data to have when deciding to use this approach. Any pH sensitivity does not detract from the utility of the sensors but having this data readily accessible to the reader would be a great help to potential users in knowing what controls to run when applying this nested FP technology.

It would be useful to quantitatively compare the Ca²⁺ responsiveness of the Matryosh sensors to e.g., the currently available FRET-based Ca²⁺ sensors to help highlight how these new sensors do represent a significant improvement in signal detection and dynamic range over the existing technology. This wouldn't necessarily need to be through new experiments and could be approached in the discussion by just comparing published FRET dynamic ranges to the Matryosh sensor data from the current paper.

Page 3 line 2. Would it be appropriate to reference one of Roger Tsien's papers here as a pioneer in the ratiometric sensor approach?

Figure 3 should show the regions of interest being quantitatively analyzed or state in the legend that these are averages over the entire field of view.

For figures 2, 5, and figure 2 suppl 1. It would be helpful to reiterate the levels of Ca²⁺ and NH₄Cl used as outlined in the methods also in the figure legends to make the color-coded traces more easily understood.

Should there be supplemental movies y and z and 1-4?

Reviewer #2, an expert in FRET-based biosensors (Remarks to the Author):

Referee report

The authors of this study describe a new design to improve several aspects of single emission, intensity based sensors. This class of biosensors employs a circular permuted GFP (cpGFP). They

owe their popularity to their high dynamic range. However, since these sensors are intensity-based (in contrast to FRET sensors, which usually provide a ratiometric read-out), quantitation is hardly possible. To solve this issue, the authors insert an inert (i.e. non-responding) fluorescent protein in the sensor, which acts as a reference. Since this reference emits at a different wavelength, the sensors can be analysed with emission ratiometric imaging.

The approach is original and promising, but several issues need to be addressed to allow publication of this work.

Major points:

1) Folding and maturation

My main worry with these type of sensors is the folding and maturation efficiency. For GFPs in general it is assumed that every protein produced will result in a fully mature fluorescent protein. In the reported sensors, the N- and C-termini of the GFP are altered, sensing domains are inserted and another fluorescent protein is added. The authors should do some effort to estimate the fraction of (folded and) mature sensor in cells.

One way to do this is to co-express a reference protein to estimate relative brightness in cells. Co-expressing eGO-Matryoshka with mCherry, and comparing it with cpGFP co-expressed with mCherry and LSSmOrange co-expressed with mCherry, will reveal whether the nested protein approach affects maturation in cells.

At this stage, where the authors show the first prototype of such a sensor, it is not important to demonstrate full maturation, but it is vital that we know how well those sensors are folded. Therefore, we need to know the fraction of functional sensor. This will allow potential users to make an independent decision to engineer new sensors based on this new design or not.

2) Quantitation

One of the advantages of ratiometric sensors is that they enable quantitation, as the authors explain in their introduction and in the discussion (p.23, lines 1-4). A first requirement is that the ratio of emission wavelengths is rather constant between cells, which can be determined for example from the data presented in figure 4. This will also provide some insight in the folding efficiency, since similar maturation efficiency of both proteins will result in equal emission intensities and hence a stable ratio, i.e. little cell-to-cell variation. However, if the proteins have different folding rates and/or maturation/efficiency, this will result in widely varying ratios of green and red fluorescence between cells.

Since the ratio of the calcium sensor is also calcium dependent, it may be worthwhile to do such a cell-based analysis of ratios between green and orange emission intensity also on the eGO-Matryoshka or sfGFP-Matryoshka.

In addition, I encourage the authors to use their ratiometric calcium sensor MatryoshCaMP6s to calculate calcium concentrations (in μM) by calibrating the ratio (as has been done previously for ratiometric calcium sensors). This would convincingly demonstrate the new potential of their sensor design.

3) In planta data

The result presented in figure 3 is promising but needs controls.

Plants roots have autofluorescence, so a non-transformed control imaged under identical circumstances is required. Secondly, it is unclear whether the observed ratio change reflects a calcium change. A pH change in the root, could provide a similar ratio change. Therefore, a non-responsive probe, e.g. eGO-Matryoshka, is an essential control.

Finally, it is unclear whether the observed result (figure 3A) is obtained from one root, or an average of multiple. I would like to see the response of at least three individual roots, which can be presented as individual curves.

4) Advantages & disadvantages

The authors state in the introduction that (ultra)violet excitation is toxic for cells (page 3 line 5). This requires a reference.

The cpGFP-LSS-mOr pair needs 440 nm excitation light which is in the violet-blue range (and identical to the optimal wavelength for excitation of FRET based sensor comprising CFP and YFP). Hence, the new design does not offer a real advantage with respect to excitation wavelength.

This could be improved by employing the CyOFP as reference, shifting excitation to 488 nm. While the authors have data with CyOFP they do not show it since the GFP-CyOFP pair gives rise to FRET (p.9, lines 1-9). I do not see why this would be a limitation and encourage the authors to show that data. The data of the calcium sensor that uses CyOFP as a reference will enable future users to decide whether it is worthwhile to test the CyOFP-cpGFP pair as well.

The authors use a large Stokes shift fluorescent protein for measuring calcium with GCaMP. Such a strategy has been demonstrated in the CyOFP paper where GCaMP was co-imaged with CyOFP expressed from a bicistronic plasmid (<https://www.ncbi.nlm.nih.gov/pubmed/27240196>). The authors should stress the advantage of their approach, which is exact co-localization between sensor and reference. Moreover, since it is a single protein, both FPs will be degraded simultaneously. In case of two proteins expressed from a bicistronic plasmid, differences in protein production and degradation will complicate the quantitation.

5) Photostability

Ratiometric read-outs are ideal for quantitation based on ratio-imaging. One of the assumptions is that the bleaching rates of both FPs are comparable, which is rare (also for FRET sensors). The authors do some effort to characterize bleaching (figure 4D). However, the experiment is performed under unrealistic high power regimes (what is the power at the objective?). The effects should be examined under realistic conditions and should the data should show both green and orange emission channels. The experiment is probably done in unstimulated cells. Since the photobleaching kinetics can be substantially different when calcium levels are high, the experiment should be performed both in a condition where calcium levels are high as well as in a condition where calcium levels are low.

6) Model

The way that I feel about the model that explains the factors contributing to the intensity change of GCaMP is as follows: either the model is an important result and is fully accommodated in the main text, or it is a side-project and should be published elsewhere. Since the model is not used to guide development of new sensors, my preference is to publish it elsewhere as a full paper.

It is unclear where the data listed in Appendix Table 1 is derived from. Please provide a reference or indicate that it was obtained in this study. I'm wondering for instance where the QY data come from?

Was it determined in this study? How? The fact that the QY of GCaMP6s differ for the apo and saturated situation is highly surprising. In fact, for cpGFP sensors it is known that the QY does not change between apo and sat (<https://www.ncbi.nlm.nih.gov/pubmed/23459413>). This can also be inferred from identical excited state kinetics (table 3:

<http://www.jneurosci.org/content/32/40/13819>).

Minor points:

1) Due to the LUT in figure 4A, it is difficult to judge expression levels and distinguish cells. Please provide these image in greyscale, which provides the best contrast.

2) When data consists of three measurements (e.g. 2C, 4C, 5C,E,F), plot the individual data, without summary statistics (mean and s.e.m.). For n=3 s.e.m is not appropriate.

3) The authors describe calcium sensor plasmid in the pDisplay backbone (page 29, line 14). The

primary goal of the pDisplay vector is to target proteins to the extracellular face of the plasma membrane as far as I know. This is somewhat confusing, since they describe that the sensors are expressed IN Hek cells (page 14, line 6). If the probes are in fact targeted to the extracellular face of the PM this leads to several follow-up questions....

4) The AmTryoshka1;3 sensor is the result of a substantial amount of engineering. Although evidence of functionality is presented (figure 5B), I would like to see cellular localization data of this membrane resident sensor in comparison to a GFP-tagged variant of the transporter. This will give insight in the correct folding, localization and functioning of the biosensor.

5) The (raw) experimental data that was used to calculate the pKa values (table 1) should be presented (graph showing intensity vs. pH), for instance in supplemental info.

Reviewer #3, an expert in theoretical photochemistry (Remarks to the Author):

As I was asked to check the mathematical model used in this paper by the editorial office, I will only focus on that part.

Basically, to my eyes, the purposes of the modeling are: (1) to show the dependence of the dynamic range to the pKa change, emissive sub-population of the apo form (f_B), and the quantum yield ratio (f_{ϕ}) between apo and saturated forms, and then (2) to suggest ways of improving the sensor performance. For this, the equilibrium model in Appendix Figure 1 is adopted. Based on this, the dynamic range is expressed with Eq 1. The physical ground of this equation is that the fraction of deprotonated (and emissive) form must be scaled by both f_B and f_{ϕ} . The model is used for guessing f_B as stated in the caption of AFig 1: as a ratio between acid/base equilibrium constants. Appendix Figures 2 & 3 and Appendix Table 1 were generated based on this model.

While I fully agree on the purposes, I sense that the authors have not fully considered one important aspect of the fluorescent protein. In the excited state, the chromophore becomes extremely acidic (some people like to call it super-photoacid) and its deprotonation is usually extremely fast as long as there is a proper partner that will accept the proton. In avGFP, E222 is known to have that partner role, while in cpFP what will take that role is not clarified yet. I am sure the authors already know this ESPT issue, because they are basing their model on Appendix Ref 5 (or Ref 27 or the main text) by Boxer's (harbinger of ESPT concept in GFP) group and because ARef 5's leading author is actually taking part in writing the present submission too. However, ARef 5 proposed the model to elucidate GROUND state proton transfer behavior. Thus, I have a strong objection to using it for characterizing emission properties.

It is possible that the excited state proton transfer is so slow and the A/B ratio in the excited state directly inherits the ratio from the ground state. If so, the present model toward generating f_B will be flawless. And some FPs are known to have blocked ESPT pathways. Is that assumption right for calcium sensing variants? I do not think there is a clear answer presently available. Remember, avGFP does not satisfy this condition and the A/B ratio in emission for avGFP (excited state property) is very different from the A/B ratio in absorption (ground state property). Unless there is some evidence that shows that the excited state composition strictly follows the ground state composition, using the model in AFig 1 toward estimating f_B will not be correct.

If this is not correct, is the publishability of the present manuscript undermined? I do not feel so. In "manufacturing" Eq 1, it is so natural to assume that the emission intensity from the apo-form should be expressed as the product of: (1) fraction f_{SS} ; (2) fraction of B state relative to the sat-form, f_B ;

(3) relative quantum yield, f_{ϕ} . Thus, even without using the model in AFig 1, we can have Eq 1 without any problem. Because dynamic ranges can be measured experimentally, and so are f_{SS} and f_{ϕ} , one can even estimate f_B from experimental data. (Though the uncertainty will be very large due to somewhat unfathomable baseline and noise issue). Thus, Eq 1 itself is not bad at all. Only the fact that AFig 1 model was used for estimating f_B is an issue. More importantly, dynamic range analysis is not a crucial part of the paper. In fact, "matryoshka'lization" of the sensor protein, namely inserting internal standard LSSmOrange to Ca-sensing cpFP, does not change the dynamic range of the original Ca-sensor anyways (Table 1). Thus, as long as the other reviewers that handled the sensor performance part of the paper feel the manuscript to be appropriate for Nature Communications, I will not argue that the limitation of the model in Appendix will be a serious enough issue. (And I indeed like the authors' idea of making dual emission sensors.) Even still, due to the issues that I raised in the above, the use of the model in AFig 1 should be revised or more evidence to justify it should be presented. In the extreme, as the manuscript is not really about improving dynamic range by nesting LSSmOrange, the authors may even consider completely removing that vague modelling part.

Another minor but important issue: there are two journals with the name Biochemistry. As far as I know, they are distinguished as "Biochemistry" and "Biochemistry (Moscow)". In the literature section, the two are widely mangled.

Reviewer #1

Addition of pH titration curves

We have added a figure with the pH titration curves of the individual cpFPs and GO-Matryoshka iterations with different N- and C-terminal residue combinations to the Supplementary Materials (Supplementary Figure 2). As illustrated, the cpFPs and GO-Matryoshka iterations show pH-sensitivity, which is additionally affected by the N- and C-terminal residues. However, the pH sensitivity is intrinsic to the cpFPs and nesting of LSSmOrange did not impact the pH-behavior as shown by similar pKa values for the GO-Matryoshka iterations (Supplementary Table 1).

Quantitative comparison of MatryoshCaMP sensors with existing FRET-based Calcium sensor

We thank the reviewer for this suggestion. Ideally we would like to perform such a comparison under the same experimental conditions with the sensors side by side, due to experimental and instrumental variations in the literature.

However, we want to state that according to a recent review (“The Growing and Glowing Toolbox of Fluorescent and Photoactive Proteins”, by Rodriguez et al, Cell Press 2017) GCaMP3 is still regarded to be the “breakthrough version” for calcium imaging. The ratiometric dynamic range of MatryoshCaMP6s is in the same range as the intensimetric GCaMP3 but with the additional benefit of a stoichiometric control. Excitation of MatryoshCaMP6s with ~488nm, as it is routinely done for GCaMP sensors, demonstrates similar properties compared to GCaMP6s which is one of the best performing calcium sensors to date.

We thus believe that the MatryoshCaMP6s sensors are state of the art sensors. Real proof of the advancement of MatryoshCaMP6s over existing calcium sensors will have to come with future applications and uses.

Reference of Roger Tsien as pioneer of ratiometric sensor approach

We have added the reference Gryniewicz, Poenie and Tsien (J Biol Chem 1985) to address Dr. Tsien's early work on synthetic dyes which exhibit excitation and emission ratiometric behavior.

Show regions of interest in Figure 3

In the Methods section, we described how we obtained these values (the ROI was defined by a binary mask made from the LSSmOrange channel), but we now also clarified this in the figure legend. We also included a supplemental movie of the binary mask that was applied (Supplementary Movie 5).

Addition of calcium and ammonium concentrations in figure legends (Figure 2, 5, Figure 2 suppl. 1)

This content was added.

Missing supplementary movies.

These are supplied with the resubmission.

Reviewer #2

1. Folding and Maturation

We understand that the fusion of sensor proteins can potentially lead to partially unfolded or immature fluorescent proteins. While we agree that this is a valid concern, the folding and maturation properties of fusion proteins can greatly vary depending on the organism and/or cell type the sensor is expressed in. Additionally, chromophores of red FPs tend to generally maintain a partially immature state (Subach and Verkhusha, Chemical Reviews 2012). Therefore we decided to take a complementary approach of optical spectroscopy and high resolution ESI-MS to understand the folding/ maturation behavior of LSSmOrange alone and nested within cpEGFP/cpsfGFP (Supplementary Figure 3, Supplementary Discussion). We indeed discovered a reduced LSSmOrange chromophore maturation in the GO-Matryoshka versions. However, we were also able to show with our mass spectrometry data analysis that the reduced LSSmOrange maturation efficiency in the GO-Matryoshkas is very similar to LSSmOrange alone and thus is most likely not due to the nesting approach but rather an intrinsic property of the FP itself.

The incomplete maturation does not limit the usefulness of the internal reference or the general Matryoshka concept because we found the fraction of immature LSSmOrange to be consistent throughout our measurements. In the future, this limitation can be overcome by choosing/designing a reporter FP with better maturation efficiency.

We refrained from the approach suggested by Reviewer #2, which would have been to co-express mCherry with eGO-Matryoshka and comparing it with cpEGFP and LSSmOrange, both co-expressed with mCherry. In our opinion, this would not be a suitable experiment to address this issue. Firstly, bicistronic plasmids have been reported to show variability among exact expression levels of the two proteins (Mizuguchi *et al*, Molecular Therapy 2000). Secondly, the overlapping spectral properties of cpEGFP and mCherry may potentially lead to FRET, which would complicate evaluation of the data. Thirdly, the premise of this manuscript is to introduce a novel approach to combine fluorescent proteins as ratiometric labels or for biosensor design. Our goal is to show that neither of the fluorescent protein properties is affected by the nesting approach, which we believe was accomplished.

2. Quantitation

There was a request for a cell-based analysis of the ratios to assess the folding rates and/or maturation efficiency. We are currently re-analyzing data from plants expressing MatryoshCaMP6s to investigate this topic; but, *a priori*, we note the inherent challenge caused by underlying calcium dynamics in live cells (i.e., variability in brightness of green emissions could be caused by differences in cytosolic calcium levels or properties / status of the fluorescent proteins themselves).

Although prior publications have reported *in vivo* calibration protocols for calcium probes, we are cautiously skeptical of the accuracy of these calibrations, particularly for plants, and do not feel it provides any meaningful information for this study. Please see the reference below for relevant discussion of this topic. This has been confirmed by pers. comm. with Gabi Monshausen, one of the pioneers of calcium imaging in plants.

Swanson, Sarah J., and Simon Gilroy. "Imaging changes in cytoplasmic calcium using the Yellow Cameleon 3.6 biosensor and confocal microscopy." *Plant Lipid Signaling Protocols* (2013): 291-302.

3. In planta data

Autofluorescence of roots of a non-transformed plant seedling was recorded under the same imaging conditions as the calcium sensor MatryoshCaMP6s in Figure 3 and the data were added to Figure 3. The illustrated traces in Figure 3 represent the result of a single root. Additional traces of individual roots were provided in Supplementary Figure 7.

It was also noted by the reviewer that a pH change in the root could lead to a similar ratio change. It has been shown that cytosolic calcium elevations in the root lead to acidification of the cytosol (Monshausen *et al*, Plant Cell. 2009), which would lead to a decrease in fluorescence intensity/ ratio as opposed to the increase of the ratio we observe. Supplementary Figure 5 illustrates the pH behavior of GCaMP6s and MatryoshCaMP6s calcium sensor iterations.

4. Advantages and disadvantages

The missing citation for UV-toxicity was added to the introduction as well as a sentence in the discussion regarding the disadvantage of the 440 nm excitation of the present Matryoshka sensors.

As suggested, we added the *in vitro* characterization (complete calcium and pH titrations) of the cyOFP1-based MatryoshCaMP6s sensor constructs and provided these data in the Supplementary Material (Supplementary Figure 6, Supplementary Table 2). The large spectral overlap of cpEGFP and cpsfGFP, respectively, with cyOFP1 allows for efficient FRET. Therefore, in the presence of calcium or any ligand, there will be two components affecting the dynamic range of the sensor: i) the green fluorescence intensity change as response to the substrate binding and ii) the change in FRET efficiency.

We thank the reviewer for pointing out so clearly the advantages of the Matryoshka technology over conventional co-expression as control. We modified the text in the discussion accordingly.

6. Photostability

The HEK293T cell assay was performed to assess whether the LSSmOrange insertion into GCaMP6s had an effect on the photobleaching properties of the cpFP. High intensity excitation was used because GCaMP6s is dim at resting stage. The transient nature of calcium spikes – they appear for a few seconds and then decay – makes it very difficult to obtain photobleaching data in the presence of calcium and to distinguish the nature of the decay.

We do agree that the photobleaching rates can be different between the saturated and apo species due to the different B-state concentrations and the quantum yield differences. However, even if the intrinsic photobleaching was the same for the saturated and apo conditions, the observed photobleaching would appear more severe for the saturated species due to its greater abundance per unit FP concentration.

It was not our intention to claim that the photobleaching properties of the two FPs are comparable. As the Reviewer mentioned, such a scenario would be rare. While obtaining absolute ratio values under imaging conditions that involve severe photobleaching might be impossible, the reference FP channel will still be a good and useful control to identify relatively well the reason for certain intensity drifts.

7. Model

The data for this model were derived from absorbance and fluorescence measurements of the sensors in this study.

Minor points

Figure 4A should be provided in greyscale.

We feel that this is a stylistic preference but can provide the greyscale images if required.

Plotting of individual data for Figure 2C, 4C, 5C, E, F

This was addressed.

pDisplay backbone

We apologize, this was a typo and was corrected in the Materials & Methods section. The mammalian expression vector used was pcDNA 3.2-Dest. We thank the reviewer for noticing this error.

Cellular localization of AmTryoshka

Yeast images were provided in Supplementary Figure 11a.

Experimental data for pKa calculation

The pH titration curves of calcium-free and calcium saturated conditions were provided in Supplementary Figure 5.

Reviewer #3

There are a number of features particular to green fluorescent proteins that justify the approach of using ground-state populations as we do even against GFP's complex excited state behavior. As noted, one of us (L.M.O.) indeed completed his Ph.D. in Prof. Steve Boxer's lab and is thus well acquainted with the ESPT issues.

Subject to the chemical environment (i.e. the pH, analyte concentration, etc.) there will be some mixture of sensor species present (i.e. apo B-state, bound A-state, and so on). The composition of the population depends on the analyte affinity as well as the acid-base characteristics of the respective apo or bound states exemplified by Appendix Fig. 1. Up to this point these are all ground-state thermodynamic parameters.

To interrogate the sensor we select a particular excitation wavelength at which we irradiate the sample and then measure the fluorescent emission at another wavelength. In our model we must determine 1) the probability an excitation photon is absorbed and 2) the probability that an excited species emits a fluorescent photon at our detection wavelength. 1) is determined by the ground-state concentration of each species as well as its extinction coefficient at the excitation wavelength. 2) can be more complex. For the simple cases of B-state species it is simply the fluorescence quantum yield. Due to ESPT the A-state presents a more complicated scenario. On picosecond timescales A* (in certain GFPs) transfers a proton to an internal terminal proton acceptor via a proton wire. Once the proton has been transferred, the chromophore is in a B*-like state which has historically been called I*. I* gives rise to fluorescence emission with some

quantum yield. In steady-state the net process (ie. $A^* \rightarrow I^* \rightarrow \text{green hv}$) has an effective quantum yield which is the product of the quantum yield of proton transfer and the fluorescence quantum yield of I^* . Therefore by knowing the extinction coefficients of all of the species and effective quantum yields for green fluorescence (either direct as for the B-state or after ESPT as for the A-state) we can determine the relative fluorescence intensity from any particular ground-state population composition.

One important detail is that we are only exciting on the B-state absorbance band. The A-state absorbance band has essentially no overlap at this wavelength. Consequently we aren't doing any A-state excitation. The acid-base driven factors of the dynamic range (the combination of the pK_a 's and the f_B parameter) lead to differences of the occupancies of the A- and B-states in which that fraction of the population in the A-state, no matter what's its fluorescence behavior, is not observed. Indeed, the differences between the A-state occupancies between the apo and bound forms of the sensors are what lead to the bulk of the dynamic response. If we were doing broadband excitation (exciting both A- and B-state bands) then we absolutely would have a big problem with dynamic range, particularly if there were significant ESPT. We have included additional text in the Appendix and have added a section to the Supplementary Materials to clarify these issues (see below).

Were we to excite only the A-state band we would indeed observe green fluorescence originating from ESPT. It is a straightforward extension of the model to encompass A-state excitation but would require the experimental determination of additional parameters (such as the effective ESPT fluorescence quantum yield). As the reviewer correctly notes, the excited-state population is emphatically not reflective of the ground-state on fluorescence lifetime timescales because the excited chromophore is indeed very acidic. The key is that we can still determine the fate of an A-state excitation, that is, the probability that A-state excitation leads ultimately to the emission of a green photon. The following equations give a generalized description of the dynamic response for arbitrary pH and excitation wavelength (λ):

Eq. S1)

$$f_{ss}(pH, pK_a) = \frac{1}{1 + 10^{(pK_a - pH)}}$$

Eq. S2)

$$F_{sat}(\lambda, pH) = f_{ss}(pH, pK_{a,sat}) * \epsilon_{B,sat}(\lambda) * \phi_{B,sat} + [1 - f_{ss}(pH, pK_{a,sat})] * \epsilon_{A,sat}(\lambda) * \phi_{ESPT,sat} * \phi_{I^*,sat} + N_0$$

Eq. S3)

$$F_{apo}(\lambda, pH) = f_{SS}(pH, pK_{a,apo}) * \varepsilon_{B,apo}(\lambda) * \phi_{B,apo} * f_B + \\ [1 - f_{SS}(pH, pK_{a,apo})] * \varepsilon_{A,apo}(\lambda) * \phi_{ESPT,apo} * \phi_{I^*,apo} + N_0$$

Eq. S4)

$$\frac{\Delta F}{F_0} = \frac{F_{sat}(\lambda, pH) - F_{apo}(\lambda, pH)}{F_{apo}(\lambda, pH) + N_0}$$

where f_{SS} is the standard single site titration sigmoid for the deprotonated species, $\varepsilon(\lambda)$ is the intrinsic extinction coefficient weighted basis spectrum for the corresponding species (e.g. apo A-state), ϕ is the quantum yield either for fluorescence or excited-state proton transfer, f_B is the internal buffering factor, and N_0 is the relative noise.

For the case we are presenting in the text we are assuming excitation only of the B-state absorbance band which causes all terms with $\varepsilon_A(\lambda)$ to be zero. Furthermore, we are assuming that $\varepsilon_{B,sat}(\lambda) = \varepsilon_{B,apo}(\lambda)$ which allows us to factor these out. Thus, the final simplified expression is given by Equation 1 in the Appendix. It was a mistake to not clearly enumerate the assumptions underlying Eq. 1 which we have now fixed. We have now included the full expressions above for the interested reader in the Supplementary Materials.

Finally, we have corrected the issue with “(Moscow)”. We thank the reviewer for their diligence in spotting this error and, more importantly, in helping improve the clarity of the dynamic range model.

Supplementary Movie Legends

Supplementary Movies 1-5 Arabidopsis seedlings expressing MatryoshCaMP6s report root cytosolic calcium elevation in response to salt shock. Scale bars, shown in upper right corner of movies, indicate 50 μ m. Elapsed time is displayed in upper left corner. Salt shock was applied at 147 seconds (s).

Supplementary Movie 1 Average z-stack projection of emission in the green channel (500-540 nm). Lookup table is shown in lower left in arbitrary units.

Supplementary Movie 2 Average z-stack projection of emission in the orange channel (570-650 nm). Lookup table is shown in lower left in arbitrary units.

Supplementary Movie 3 Ratio of green-to-orange emission in average z-stack projections after application of a binary mask, shown in SM5. Lookup table is shown in lower left corner, with ratio values listed.

Supplementary Movies 4 Transmitted light (Brightfield), single-plane only.

Supplementary Movies 5 Binary mask generated from projection of orange channel and applied to ratiometric data shown in SM3.

Thank you very much

Wolf Frommer

REVIEWERS' COMMENTS:

Reviewer #1 (Remarks to the Author):

The authors have done a great job of answering the queries raised in the review. The extra in vitro characterizations especially will be very useful for potential users to see the strengths of these new probes that should provide important additions to the available in vivo biosensor toolkit.

Reviewer #2 (Remarks to the Author):

The revised manuscript is a substantial improvement. I have a few issues that need to be addressed to improve the manuscript. The main point is that the discussion should give a fair impression to anyone that wants to adopt this approach as to what issues may be encountered when implementing this technology.

1) Folding and Maturation

The authors have examined the folding of GFP and the internal control (LSS-mOrange). I think the authors should explicitly state the folding ratio that they experimentally determined in the main text. In addition, it needs to be discussed that these values are obtained from protein purified from E.coli and that, as a consequence, the values in plant and mammalian cells can be different.

2) Quantitation

If the authors are "cautiously skeptical" of the accuracy of calibrations, this should be discussed in the paper. The authors may indicate that these sensors potentially allow quantitation, if accurate calibration procedures are available.

3) In planta data

It would be useful to know the level of autofluorescence intensity with respect to the level of GFP fluorescence. Images of an autofluorescent root and a transformed root under the same conditions would be sufficient.

With respect to the pH changes, the reference to Monshausen is very useful and should be included in the paper.

4) Advantages and disadvantages

I fully agree with the response and modifications of the authors

6) Photostability

It would be useful to add a bit of discussion about the photostability to the discussion

In general, it would be good to add to the discussion how this new probe can be used for quantitation and what possible challenges lie ahead (i.e. maturation, photostability of the two probes, calibration measurements). This would give new users a fair idea of what is necessary to successfully implement the (promising!) technology.

7) Model

My previous remark about the model is not addressed: "The way that I feel about the model that explains the factors contributing to the intensity change of GCaMP is as follows: either the model is an important result and is fully accommodated in the main text, or it is a side-project and should be published elsewhere. Since the model is not used to guide development of new sensors, my preference is to publish it elsewhere as a full paper."

But if none of the other referees agrees with me and the authors want to leave it as is, I rest my case .

8) The authors should add a reference to a paper in which a similar strategy was used (this does not conflict with the novelty of the current paper):

<http://dx.doi.org/10.1021/acscchembio.6b00883>

Reviewer #3 (Remarks to the Author):

The authors have a lot more clearly explained the assumptions of their model, and the revisions they took along the course make me feel that the manuscript is now ready for publication without any further revisions.

Reviewer #1

It would be very useful to add empirical pH titration curves. There is a lot of pH dynamic analysis in the Appendix, but the cpGFPs are often very pH sensitive and effects of pH on each signal peak intensities will be important data to have when deciding to use this approach. Any pH sensitivity does not detract from the utility of the sensors but having this data readily accessible to the reader would be a great help to potential users in knowing what controls to run when applying this nested FP technology.

We have added a figure with the pH titration curves of the individual cpFPs and GO-Matryoshka iterations with different N- and C-terminal residue combinations to the Supplementary Materials (Supplementary Figure 2). As illustrated, the cpFPs and GO-Matryoshka iterations show pH-sensitivity, which is additionally affected by the N- and C-terminal residues. However, the pH sensitivity is intrinsic to the cpFPs and nesting of LSSmOrange did not impact the pH-behavior as shown by similar pKa values for the GO-Matryoshka iterations (Supplementary Table 1).

It would be useful to quantitatively compare the Ca²⁺ responsiveness of the Matryosh sensors to e.g., the currently available FRET-based Ca²⁺ sensors to help highlight how these new sensors do represent a significant improvement in signal detection and dynamic range over the existing technology. This wouldn't necessarily need to be through new experiments and could be approached in the discussion by just comparing published FRET dynamic ranges to the Matryosh sensor data from the current paper.

We thank the reviewer for this suggestion. Ideally, we would like to perform such a comparison under the same experimental conditions with the sensors side by side, due to experimental and instrumental variations in the literature.

However, we want to state that according to a recent review (“The Growing and Glowing Toolbox of Fluorescent and Photoactive Proteins”, by Rodriguez et al, Cell Press 2017) GCaMP3 is still regarded to be the “breakthrough version” for calcium imaging. The ratiometric dynamic range of MatryoshCaMP6s is in the same range as the intensimetric GCaMP3 but with the additional benefit of a stoichiometric control. Excitation of MatryoshCaMP6s with ~488nm, as it is routinely done for GCaMP sensors, demonstrates similar properties compared to GCaMP6s which is one of the best performing calcium sensors to date.

We thus believe that the MatryoshCaMP6s sensors are state of the art sensors. Real proof of the advancement of MatryoshCaMP6s over existing calcium sensors will have to come with future applications and uses.

Page 3 line 2. Would it be appropriate to reference one of Roger Tsien's papers here as a pioneer in the ratiometric sensor approach?

We have added the reference Grynkiewicz, Poenie and Tsien (J Biol Chem 1985) to address Dr. Tsien's early work on synthetic dyes which exhibit excitation and emission ratiometric behavior.

Figure 3 should show the regions of interest being quantitatively analyzed or state in the legend that these are averages over the entire field of view.

In the Methods section, we described how we obtained these values (the ROI was defined by a binary mask made from the LSSmOrange channel), but we now also clarified this in the figure legend. We also included a supplemental movie of the binary mask that was applied (Supplementary Movie 5).

For figures 2, 5, and figure 2 suppl 1. It would be helpful to reiterate the levels of Ca²⁺ and NH₄Cl used as outlined in the methods also in the figure legends to make the color-coded traces more easily understood.

This content was added.

Should there be supplemental movies y and z and 1-4?

These are supplied with the resubmission.

Reviewer #2 (first submission)

1) Folding and maturation

My main worry with these type of sensors is the folding and maturation efficiency. For GFPs in general it is assumed that every protein produced will result in a fully mature fluorescent protein. In the reported sensors, the N- and C-termini of the GFP are altered, sensing domains are inserted and another fluorescent protein is added. The authors should do some effort to estimate the fraction of (folded and) mature sensor in cells.

One way to do this is to co-express a reference protein to estimate relative brightness in cells. Co-expressing eGO-Matryoshka with mCherry, and comparing it with cpGFP co-expressed with mCherry and LSSmOrange co-expressed with mCherry, will reveal whether the nested protein approach affects maturation in cells.

At this stage, where the authors show the first prototype of such a sensor, it is not important to demonstrate full maturation, but it is vital that we know how well those sensors are folded. Therefore, we need to know the fraction of functional sensor. This will allow potential users to make an independent decision to engineer new sensors based on this new design or not.

We understand that the fusion of sensor proteins can potentially lead to partially unfolded or immature fluorescent proteins. While we agree that this is a valid concern, the folding and maturation properties of fusion proteins can greatly vary depending on the organism and/or cell type the sensor is expressed in. Additionally, chromophores of red FPs tend to generally maintain a partially immature state (Subach and Verkhusha, Chemical Reviews 2012). Therefore, we decided to take a complementary approach of optical spectroscopy and high resolution ESI-MS to understand the folding/ maturation behavior of LSSmOrange alone and nested within cpEGFP/cpsfGFP (Supplementary Figure 3, Supplementary Discussion). We indeed discovered a reduced LSSmOrange chromophore maturation in the GO-Matryoshka versions. However, we were also able to show with our mass spectrometry data analysis that the reduced LSSmOrange maturation efficiency in the GO-Matryoshkas is very similar to LSSmOrange alone and thus is most likely not due to the nesting approach but rather an intrinsic property of the FP itself.

The incomplete maturation does not limit the usefulness of the internal reference or the general Matryoshka concept because we found the fraction of immature LSSmOrange to be consistent throughout our measurements. In the future, this limitation can be overcome by choosing/designing a reporter FP with better maturation efficiency.

We refrained from the approach suggested by Reviewer #2, which would have been to co-express mCherry with eGO-Matryoshka and comparing it with cpEGFP and LSSmOrange, both co-expressed with mCherry. In our opinion, this would not be a suitable experiment to address this issue. Firstly, bicistronic plasmids have been reported to show variability among exact expression levels of the two proteins (Mizuguchi *et al*, Molecular Therapy 2000). Secondly, the overlapping spectral properties of cpEGFP and mCherry may potentially lead to FRET, which would complicate evaluation of the data. Thirdly, the premise of this manuscript is to introduce a novel approach to combine fluorescent proteins as ratiometric labels or for biosensor design. Our goal is to show that neither of the fluorescent protein properties is affected by the nesting approach, which we believe was accomplished.

2) Quantitation

One of the advantages of ratiometric sensors is that they enable quantitation, as the authors explain in their introduction and in the discussion (p.23, lines 1-4). A first requirement is that the ratio of emission wavelengths is rather constant between cells, which can be determined for example from the data presented in figure 4. This will also provide some insight in the folding efficiency, since similar maturation efficiency of both proteins will result in equal emission intensities and hence a stable ratio, i.e. little cell-to-cell variation. However, if the proteins have different folding rates and/or maturation/efficiency, this will result in widely varying ratios of green and red fluorescence between cells.

Since the ratio of the calcium sensor is also calcium dependent, it may be worthwhile to do such a cell-based analysis of ratios between green and orange emission intensity also on the eGO-Matryoshka or sfGFP-Matryoshka.

In addition, I encourage the authors to use their ratiometric calcium sensor MatryoshCaMP6s to calculate calcium concentrations (in μM) by calibrating the ratio (as has been done previously for ratiometric calcium sensors). This would convincingly demonstrate the new potential of their sensor design.

There was a request for a cell-based analysis of the ratios to assess the folding rates and/or maturation efficiency. We are currently re-analyzing data from plants expressing MatryoshCaMP6s to investigate this topic; but, *a priori*, we note the inherent challenge caused by underlying calcium dynamics in live cells (i.e., variability in brightness of green emissions could be caused by differences in cytosolic calcium levels or properties / status of the fluorescent proteins themselves).

Although prior publications have reported *in vivo* calibration protocols for calcium probes, we are cautiously skeptical of the accuracy of these calibrations, particularly for plants, and do not feel it provides any meaningful information for this study. Please see the reference below for relevant discussion of this topic. This has been confirmed by pers. comm. with Gabi Monshausen, one of the pioneers of calcium imaging in plants.

Swanson, Sarah J., and Simon Gilroy. "Imaging changes in cytoplasmic calcium using the Yellow Cameleon 3.6 biosensor and confocal microscopy." *Plant Lipid Signaling Protocols* (2013): 291-302.

3) In planta data

The result presented in figure 3 is promising but needs controls.

Plants roots have autofluorescence, so a non-transformed control imaged under identical circumstances is required. Secondly, it is unclear whether the observed ratio change reflects a calcium change. A pH change in the root, could provide a similar ratio change. Therefore, a non-responsive probe, e.g. eGO-Matryoshka, is an essential control.

Finally, it is unclear whether the observed result (figure 3A) is obtained from one root, or an average of multiple. I would like to see the response of at least three individual roots, which can be presented as individual curves.

Autofluorescence of roots of a non-transformed plant seedling was recorded under the same imaging conditions as the calcium sensor MatryoshCaMP6s in Figure 3 and the data were added to Figure 3. The illustrated traces in Figure 3 represent the result of a single root. Additional traces of individual roots were provided in Supplementary Figure 7.

It was also noted by the reviewer that a pH change in the root could lead to a similar ratio change. It has been shown that cytosolic calcium elevations in the root lead to acidification of the cytosol (Monshausen et al, Plant Cell. 2009), which would lead to a decrease in fluorescence intensity/ ratio as opposed to the increase of the ratio we observe. Supplementary Figure 5 illustrates the pH behavior of GCaMP6s and MatryoshCaMP6s calcium sensor iterations.

4) Advantages & disadvantages

The authors state in the introduction that (ultra)violet excitation is toxic for cells (page 3 line 5). This requires a reference.

The cpGFP-LSS-mOr pair needs 440 nm excitation light which is in the violet-blue range (and identical to the optimal wavelength for excitation of FRET based sensor comprising CFP and YFP). Hence, the new design does not offer a real advantage with respect to excitation wavelength.

This could be improved by employing the CyOFP as reference, shifting excitation to 488 nm. While the authors have data with CyOFP they do not show it since the GFP-CyOFP pair gives rise to FRET (p.9, lines 1-9). I do not see why this would be a limitation and encourage the authors to show that data. The data of the calcium sensor that uses CyOFP as a reference will enable future users to decide whether it is worthwhile to test the CyOFP-cpGFP pair as well.

The authors use a large Stokes shift fluorescent protein for measuring calcium with GCaMP. Such a strategy has been demonstrated in the CyOFP paper where GCaMP was co-imaged with CyOFP expressed from a bicistronic plasmid (<https://www.ncbi.nlm.nih.gov/pubmed/27240196>). The authors should stress the advantage of their approach, which is exact co-localization between sensor and reference. Moreover, since it is a single protein, both FPs will be degraded simultaneously. In case of two proteins expressed from a bicistronic plasmid, differences in protein production and degradation will complicate the quantitation.

The missing citation for UV-toxicity was added to the introduction as well as a sentence in the discussion regarding the disadvantage of the 440 nm excitation of the present Matryoshka sensors.

As suggested, we added the *in vitro* characterization (complete calcium and pH titrations) of the cyOFP1-based MatryoshCaMP6s sensor constructs and provided these data in the Supplementary Material (Supplementary Figure 6, Supplementary Table 2). The large spectral overlap of cpEGFP and cpsfGFP, respectively, with cyOFP1 allows for efficient FRET. Therefore, in the presence of calcium or any ligand, there will be two components affecting the dynamic range of the sensor: i) the green fluorescence intensity change as response to the substrate binding and ii) the change in FRET efficiency.

We thank the reviewer for pointing out so clearly the advantages of the Matryoshka technology over conventional co-expression as control. We modified the text in the discussion accordingly.

5) Photostability

Ratiometric read-outs are ideal for quantitation based on ratio-imaging. One of the assumptions is that the bleaching rates of both FPs are comparable, which is rare (also for FRET sensors). The authors do some effort to characterize bleaching (figure 4D). However, the experiment is performed under unrealistic high power regimes (what is the power at the objective?). The effects should be examined under realistic conditions and should the data should show both green and orange emission channels.

The experiment is probably done in unstimulated cells. Since the photobleaching kinetics can be substantially different when calcium levels are high, the experiment should be performed both in a condition where calcium levels are high as well as in a condition where calcium levels are low.

The HEK293T cell assay was performed to assess whether the LSSmOrange insertion into GCaMP6s had an effect on the photobleaching properties of the cpFP. High intensity excitation was used because GCaMP6s is dim at resting stage. The transient nature of calcium spikes – they appear for a few seconds and then decay – makes it very difficult to obtain photobleaching data in the presence of calcium and to distinguish the nature of the decay.

We do agree that the photobleaching rates can be different between the saturated and apo species due to the different B-state concentrations and the quantum yield differences. However, even if the intrinsic photobleaching was the same for the saturated and apo conditions, the observed photobleaching would appear more severe for the saturated species due to its greater abundance per unit FP concentration.

It was not our intention to claim that the photobleaching properties of the two FPs are comparable. As the Reviewer mentioned, such a scenario would be rare. While obtaining absolute ratio values under imaging conditions that involve severe photobleaching might be impossible, the reference FP channel will still be a good and useful control to identify relatively well the reason for certain intensity drifts.

6) Model

The way that I feel about the model that explains the factors contributing to the intensity change of GCaMP is as follows: either the model is a important result and is fully accomodated in the

main text, or it is a side-project and should be published elsewhere. Since the model is not used to guide development of new sensors, my preference is to publish it elsewhere as a full paper.

It is unclear where the data listed in Appendix Table 1 is derived from. Please provide a reference or indicate that it was obtained in this study. I'm wondering for instance where the QY data come from? Was it determined in this study? How? The fact that the QY of GCaMP6s differ for the apo and saturated situation is highly surprising. In fact, for cpGFP sensors it is known that the QY does not change between apo and sat (<https://www.ncbi.nlm.nih.gov/pubmed/23459413>). This can also be inferred from identical excited state kinetics (table 3: <http://www.jneurosci.org/content/32/40/13819>).

The data for this model were derived from absorbance and fluorescence measurements of the sensors in this study.

Minor points

1) Due to the LUT in figure 4A, it is difficult to judge expression levels and distinguish cells. Please provide these image in greyscale, which provides the best contrast

We feel that this a stylistic preference but can provide the greyscale images if required.

2) When data consists of three measurements (e.g. 2C, 4C, 5C, E, F), plot the individual data, without summary statistics (mean and s.e.m.). For n=3 s.e.m is not appropriate.

This was addressed.

3) The authors describe calcium sensor plasmid in the pDisplay backbone (page 29, line 14). The primary goal of the pDisplay vector is to target proteins to the extracellular face of the plasma membrane as far as I know. This is somewhat confusing, since they describe that the sensors are expressed IN Hek cells (page 14, line 6). If the probes are in fact targeted to the extracellular face of the PM this leads to several follow-up questions....

We apologize, this was a typo and was corrected in the Materials & Methods section. The mammalian expression vector used was pcDNA 3.2-Dest. We thank the reviewer for noticing this error.

4) The AmTryoshka1;3 sensor is the result of a substantial amount of engineering. Although evidence of functionality is presented (figure 5B), I would like to see cellular localization data of this membrane resident sensor in comparison to a GFP-tagged variant of the transporter. This will give insight in the correct folding, localization and functioning of the biosensor.

Yeast images were provided in Supplementary Figure 11a.

5) The (raw) experimental data that was used to calculate the pKa values (table 1) should be presented (graph showing intensity vs. pH), for instance in supplemental info.

The pH titration curves of calcium-free and calcium saturated conditions were provided in Supplementary Figure 5.

Reviewer #2 (second submission)

The revised manuscript is a substantial improvement. I have a few issues that need to be addressed to improve the manuscript. The main point is that the discussion should give a fair impression to anyone that wants to adopt this approach as to what issues may be encountered when implementing this technology.

1) Folding and Maturation

The authors have examined the folding of GFP and the internal control (LSS-mOrange). I think the authors should explicitly state the folding ratio that they experimentally determined in the main text. In addition, it needs to be discussed that these values are obtained from protein purified from E.coli and that, as a consequence, the values in plant and mammalian cells can be different.

The ratio values determined in our analysis were added to the main text as well as sentence regarding the potential differences of chromophore maturation when other biological systems are used.

2) Quantitation

If the authors are "cautiously skeptical" of the accuracy of calibrations, this should be discussed in the paper. The authors may indicate that these sensors potentially allow quantitation, if accurate calibration procedures are available.

This limitation was added to the discussion.

3) In planta data

It would be useful to know the level of autofluorescence intensity with respect to the level of GFP fluorescence. Images of an autofluorescent root and a transformed root under the same conditions would be sufficient.

With respect to the pH changes, the reference to Monshausen is very useful and should be included in the paper.

Confocal images of a control seedling root in comparison to seedlings expressing the sensor imaged under the same conditions were added to the supplement (Supplementary Fig. 7b).

Regarding the pH changes, a sentence of cytosolic acidification upon elevated calcium levels in the cytosol and the reference to Monshausen et al, 2009 was added to the main text.

4) Advantages and disadvantages

I fully agree with the response and modifications of the authors

5) Photostability

It would be useful to add a bit of discussion about the photostability to the discussion. In general, it would be good to add to the discussion how this new probe can be used for quantitation and what possible challenges lie ahead (i.e. maturation, photostability of the two probes, calibration measurements). This would give new users a fair idea of what is necessary to successfully implement the (promising!) technology.

We added a paragraph that discusses the potential limitation of the presented Matryoshka combinations at the end of the discussion.

6) Model

My previous remark about the model is not addressed: "The way that I feel about the model that explains the factors contributing to the intensity change of GCaMP is as follows: either the model is an important result and is fully accommodated in the main text, or it is a side-project and should be published elsewhere. Since the model is not used to guide development of new sensors, my preference is to publish it elsewhere as a full paper."

But if none of the other referees agrees with me and the authors want to leave it as is, I rest my case.

8) *The authors should add a reference to a paper in which a similar strategy was used (this does not conflict with the novelty of the current paper): <http://dx.doi.org/10.1021/acscchembio.6b00883>*

The recommended reference was added to the introduction.

Reviewer #3

There are a number of features particular to green fluorescent proteins that justify the approach of using ground-state populations as we do even against GFP's complex excited state behavior. As noted, one of us (L.M.O.) indeed completed his Ph.D. in Prof. Steve Boxer's lab and is thus well acquainted with the ESPT issues.

Subject to the chemical environment (i.e. the pH, analyte concentration, etc.) there will be some mixture of sensor species present (i.e. apo B-state, bound A-state, and so on). The composition of the population depends on the analyte affinity as well as the acid-base characteristics of the respective apo or bound states exemplified by Supplementary Fig. 13. Up to this point these are all ground-state thermodynamic parameters.

To interrogate the sensor we select a particular excitation wavelength at which we irradiate the sample and then measure the fluorescent emission at another wavelength. In our model we must determine 1) the probability an excitation photon is absorbed and 2) the probability that an excited species emits a fluorescent photon at our detection wavelength. 1) is determined by the ground-state concentration of each species as well as its extinction coefficient at the excitation

wavelength. 2) can be more complex. For the simple cases of B-state species it is simply the fluorescence quantum yield. Due to ESPT the A-state presents a more complicated scenario. On picosecond timescales A* (in certain GFPs) transfers a proton to an internal terminal proton acceptor via a proton wire. Once the proton has been transferred, the chromophore is in a B*-like state which has historically been called I*. I* gives rise to fluorescence emission with some quantum yield. In steady-state the net process (ie. A* → I* → green hv) has an effective quantum yield which is the product of the quantum yield of proton transfer and the fluorescence quantum yield of I*. Therefore by knowing the extinction coefficients of all of the species and effective quantum yields for green fluorescence (either direct as for the B-state or after ESPT as for the A-state) we can determine the relative fluorescence intensity from any particular ground-state population composition.

One important detail is that we are only exciting on the B-state absorbance band. The A-state absorbance band has essentially no overlap at this wavelength. Consequently we aren't doing any A-state excitation. The acid-base driven factors of the dynamic range (the combination of the pK_a's and the f_B parameter) lead to differences of the occupancies of the A- and B-states in which that fraction of the population in the A-state, no matter what's its fluorescence behavior, is not observed. Indeed, the differences between the A-state occupancies between the apo and bound forms of the sensors are what lead to the bulk of the dynamic response. If we were doing broadband excitation (exciting both A- and B-state bands) then we absolutely would have a big problem with dynamic range, particularly if there were significant ESPT. We have included additional text in Supplementary Note 2 and have added a section to the Supplementary Materials to clarify these issues (see below).

Were we to excite only the A-state band we would indeed observe green fluorescence originating from ESPT. It is a straightforward extension of the model to encompass A-state excitation but would require the experimental determination of additional parameters (such as the effective ESPT fluorescence quantum yield). As the reviewer correctly notes, the excited-state population is emphatically not reflective of the ground-state on fluorescence lifetime timescales because the excited chromophore is indeed very acidic. The key is that we can still determine the fate of an A-state excitation, that is, the probability that A-state excitation leads ultimately to the emission of a green photon. The following equations give a generalized description of the dynamic response for arbitrary pH and excitation wavelength (λ):

Eq. S1)

$$f_{ss}(pH, pK_a) = \frac{1}{1 + 10^{(pK_a - pH)}}$$

Eq. S2)

$$F_{sat}(\lambda, pH) = f_{ss}(pH, pK_{a,sat}) * \epsilon_{B,sat}(\lambda) * \phi_{B,sat} + [1 - f_{ss}(pH, pK_{a,sat})] * \epsilon_{A,sat}(\lambda) * \phi_{ESPT,sat} * \phi_{I^*,sat} + N_0$$

Eq. S3)

$$F_{apo}(\lambda, pH) = f_{SS}(pH, pK_{a,apo}) * \varepsilon_{B,apo}(\lambda) * \phi_{B,apo} * f_B + \\ [1 - f_{SS}(pH, pK_{a,apo})] * \varepsilon_{A,apo}(\lambda) * \phi_{ESPT,apo} * \phi_{I^*,apo} + N_0$$

Eq. S4)

$$\frac{\Delta F}{F_0} = \frac{F_{sat}(\lambda, pH) - F_{apo}(\lambda, pH)}{F_{apo}(\lambda, pH) + N_0}$$

where f_{ss} is the standard single site titration sigmoid for the deprotonated species, $\varepsilon(\lambda)$ is the intrinsic extinction coefficient weighted basis spectrum for the corresponding species (e.g. apo A-state), ϕ is the quantum yield either for fluorescence or excited-state proton transfer, f_B is the internal buffering factor, and N_0 is the relative noise.

For the case we are presenting in the text we are assuming excitation only of the B-state absorbance band which causes all terms with $\varepsilon_A(\lambda)$ to be zero. Furthermore, we are assuming that $\varepsilon_{B,sat}(\lambda) = \varepsilon_{B,apo}(\lambda)$ which allows us to factor these out. Thus, the final simplified expression is given by Equation 1 in Supplementary Note 2. It was a mistake to not clearly enumerate the assumptions underlying Eq. 1 which we have now fixed. We have now included the full expressions above for the interested reader in the Supplementary Materials.

Finally, we have corrected the issue with “(Moscow)”. We thank the reviewer for their diligence in spotting this error and, more importantly, in helping improve the clarity of the dynamic range model.